# LINE-1 RNA triggers matrix formation in bone cells via a PKR-mediated inflammatory response

Arianna Mangiavacchi [1✉], Gabriele Morelli [1], Sjur Reppe [2,3,4], Alfonso Saera-Vila[5], Peng Liu[1], Benjamin Eggerschwiler[6,7], Huoming Zhang[8], Dalila Bensaddek[8], Elisa A Casanova[6], Carolina Medina Gomez[9], Vid Prijatelj [9], Francesco Della Valle [1,10], Nazerke Atinbayeva [1], Juan Carlos Izpisua Belmonte[10], Fernando Rivadeneira[9], Paolo Cinelli [6,11], Kaare Morten Gautvik [3,12] & Valerio Orlando [1,12✉]

## Abstract

**Transposable elements (TEs) are mobile genetic modules of viral derivation that have been co-opted to become modulators of mammalian gene expression. TEs are a major source of endogenous dsRNAs, signaling molecules able to coordinate inflammatory responses in various physiological processes. Here, we provide evidence for a positive involvement of TEs in inflammation-driven bone repair and mineralization. In newly fractured mice bone, we observed an early transient upregulation of repeats occurring concurrently with the initiation of the inflammatory stage. In human bone biopsies, analysis revealed a significant correlation between repeats expression, mechanical stress and bone mineral density. We investigated a potential link between LINE-1 (L1) expression and bone mineralization by delivering a synthetic L1 RNA to osteoporotic patient-derived mesenchymal stem cells and observed a dsRNA-triggered protein kinase (PKR)-mediated stress response that led to strongly increased mineralization. This response was associated with a strong and transient inflammation, accompanied by a global translation attenuation induced by eIF2α phosphorylation. We demonstrated that L1 transfection reshaped the secretory profile of osteoblasts, triggering a paracrine activity that stimulated the mineralization of recipient cells.**

**Keywords** dsRNA; Inflammation; Osteoblast; PKR; Transposable Elements
**Subject Categories** Molecular Biology of Disease; Signal Transduction

## Introduction

Repetitive elements are, in origin, parasitic transposable elements (TEs) of viral derivation and, during evolution, have become regulatory modules of gene expression, eventually integrated into developmental, resilience, and cell identity programs (Cosby et al, 2019; Mangiavacchi et al, 2021; Jachowicz et al, 2017). Although cells have evolved several defense mechanisms to prevent deleterious uncontrolled retrotransposon reactivation (Slotkin and Martienssen, 2007), increasing evidence demonstrates that retrotransposons are also involved in non-pathological contexts, particularly as non-coding RNAs (Mangiavacchi et al, 2021). As part of resilience mechanisms, all retrotransposons are a major source of endogenous double-stranded RNAs (dsRNAs), that serve as a cellular signaling molecule to coordinate inflammatory and immune responses in various physiological processes (Fukuda et al, 2021; De Cecco et al, 2019a; Simon et al, 2019; Sadeq et al, 2021; Chen and Hur, 2022; Zhang et al, 2020). The transient and controlled breaching of the activation threshold of dsRNA sensors allows the integration of innate immune functions within the frame of physiological processes (Chen and Hur, 2022). For example, repetitive elements transcribed during development drive RIG-I-like receptors (RLR)-mediated inflammatory signals that regulate hematopoietic stem and progenitor cells (HSPC) formation (Lefkopoulos et al, 2020). During cell replication, dsRNA-mediated activation of PKR ensures proper progression of mitosis (Kim et al, 2014). Moreover, epigenetically mediated derepression of repetitive elements leads to dsRNA production and to the activation of an inflammatory response stimulating anti-tumor T cell immunity (Sheng et al, 2018). Furthermore, after a skin injury, dsRNAs induce inflammatory pathways contributing to wound healing and hair regeneration (Nelson et al, 2015). Several studies have demonstrated a positive effect of proinflammatory mediators on bone anabolism in vitro and in vivo (Croes et al, 2015; Croes

[1]King Abdullah University of Science and Technology (KAUST), Biological Environmental Science and Engineering Division, Thuwal 23500-6900, Kingdom of Saudi Arabia. [2]Oslo University Hospital, Department of Medical Biochemistry, Oslo, Norway. [3]Lovisenberg Diaconal Hospital, Unger-Vetlesen Institute, Oslo, Norway. [4]Oslo University Hospital, Department of Plastic and Reconstructive Surgery, Oslo, Norway. [5]Sequentia Biotech, Carrer Comte D'Urgell 240, Barcelona 08036, Spain. [6]Department of Trauma, University Hospital Zurich, Sternwartstrasse 14, 8091 Zurich, Switzerland. [7]Life Science Zurich Graduate School, University of Zurich, Winterthurerstrasse 190, 8057 Zurich, Switzerland. [8]Core Labs, King Abdullah University of Science and Technology (KAUST), Thuwal 23500-6900, Kingdom of Saudi Arabia. [9]Department of Internal Medicine, Erasmus Medical Centre, Rotterdam, the Netherlands. [10]Altos Labs, San Diego, CA, USA. [11]Center for Applied Biotechnology and Molecular Medicine, University of Zurich, Winterthurerstrasse 190, 8057 Zurich, Switzerland. [12]These authors contributed equally: Kaare Morten Gautvik, Valerio Orlando. ✉E-mail: Arianna.mangiavacchi@kaust.edu.sa; Valerio.orlando@kaust.edu.sa

et al, 2016; Li et al, 2016; Croes et al, 2017; Laroche et al, 2011), suggesting that an inflammatory reaction may be harnessed for bone regenerative purposes. Indeed, sterile inflammation is a fundamental component of the osteogenic microenvironment: the inflammatory reaction is the earliest response to injury and is crucial to initiate and orchestrate fracture repair and activate bone anabolic processes (Mountziaris et al, 2011; Marsell and Einhorn, 2011; Bahney et al, 2019). Accordingly, a dysregulated inflammation negatively impacts optimal bone regeneration (Wheatley et al, 2019; Gerstenfeld et al, 2001; Recknagel et al, 2013; Hurtgen et al, 2016; Jiao et al, 2015), and often precedes excessive bone formation associated with heterotopic ossification and several vertebral column pathological conditions (Lories and Schett, 2012; Balboni et al, 2006). Therefore, identifying the key molecular factors triggering the sterile inflammation that precedes regeneration in bone would allow us to modulate the inflammatory reaction in a spatiotemporal and contextual manner as well as to develop new anabolic strategies for the treatment of bone loss conditions, such as osteoporosis, or impaired bone repair (Roberts and Ke, 2018).

Here, we show that (1) In mice, immediately after bone fracture, a transient upregulation of TEs concurs with the initiation of the inflammatory stage; (2) In humans, an increased expression of repeats is observed in bones with a higher mechanical stress-induced anabolic demand and expression of TEs correlates with local mineral density; (3) The delivery of L1 RNA, but not of a control RNA, to human bone marrow-derived mesenchymal stem cells committed to osteoblasts stimulates a unique mineralizing phenotype in a dose-dependent manner; (4) L1-treated osteoblasts show upregulation of inflammatory genes and transcriptional changes characteristic for the earliest stage of bone repair; (5) Cytoplasmic L1 RNA is sensed by dsRNA-activated protein kinase R (PKR), which mediates eukaryotic translation initiation factor 2 alpha (eIF2$\alpha$) phosphorylation and a global attenuation of protein synthesis; (6) The inhibition of PKR prevents inflammation, translation inibition and mineralization caused by elevated levels of cytoplasmic L1 RNA; (7) Cells transfected with L1 RNA undergo significant changes in their secretome composition and initiate a paracrine effect stimulating the mineralization of osteogenic competent recipient cells.

Our results indicate that TEs are induced by bone damage and stimulate bone mineralization through an inflammatory response triggered by dsRNA sensing and mediated by paracrine activity.

# Results

## Retrotransposon expression is triggered by fracture in vivo and correlates with bone mineral density in human weight-bearing bones

Inflammation is the earliest response to fracture. To investigate a possible involvement of TEs in the sterile inflammatory response triggered by a bone injury, we analyzed TEs expression in time-course RNA-Seq repository data obtained from a full-fracture bone healing mice model (Coates et al, 2019). We focused on TEs expression dynamics immediately after fracture (acute inflammatory stage) and observed four clusters of differentially expressed TEs (logFC >0.5) between intact and post-fracture bone after 4 h, a time point corresponding to the initiation of acute inflammation

(Fig. 1A). Cluster 1 includes TEs subfamilies whose expression slightly increases after fracture and during the whole healing process. Clusters 2 and 4 include TEs subfamilies that are downregulated after fracture but show increased expression at later time points. The major group, cluster 3, is represented by TEs subfamilies whose expression is highly and transiently upregulated after fracture (Fig. 1A,B). Interestingly, the induction of these TEs, mostly long interspersed nuclear elements (LINEs) and long terminal repeats (LTRs) (Fig. 1B), is transient and limited to the earliest phase of the inflammatory stage necessary and sufficient to initiate and orchestrate fracture repair and activate bone anabolic processes (Mountziaris et al, 2011; Marsell and Einhorn, 2011). These results suggest that, in bone, retrotransposon reactivation is an early event in response to fracture, possibly linked to inflammation initiation which is fundamental for a healthy healing. Indeed, retrotransposon involvement in posttraumatic tissue regeneration was previously found in other organisms (Mashanov et al, 2012; Zhu et al, 2012).

To assess gene expression dynamics immediately after fracture in humans represents a technical and ethical hurdle. A comparison between bones experiencing different degrees of mechanical loading may represent an alternative model to study stress-induced bone anabolism in humans. Mechanically loaded bones like the femur, which are subject to recurring microfractures that need to be repaired, are indicated to be more metabolically active than less loaded bones, like the iliac crest, as pointed out by previous transcriptome studies (Aerssens et al, 1997; Varanasi et al, 2010).

We compared the expression of TEs in trabecular bone from the ilium and femoral head, two different skeletal sites, the latter experiencing a higher degree of mechanical loading, anabolic demand and bone turnover (Aerssens et al, 1997). Femur ($n = 48$) and ilium biopsies ($n = 71$) were isolated from a cohort of clinically well-characterized donors (refer to cohort description and Appendix Tables S1, S2). Each cohort was divided into normal (BMD T-score $> -1$), osteopenic ($-2.5 <$ BMD T-score $\leq -1$), and osteoporotic (BMD T-score $\leq -2.5$, with at least one fragility fracture). TEs show generally higher upregulation in healthy femurs ($n = 27$) compared to healthy ilia ($n = 34$) (Fig. 1C,E). Each TEs order showed a high percentage of upregulated subfamilies, from 60% of DNA transposon to 75% of LINEs, and a low percentage of downregulated subfamilies, from 9% of LTRs to 3% of short interspersed nuclear elements (SINEs) and LINEs, in femur compared to ilium (Fig. 1D). These results suggested a positive correlation between TEs global expression and bone anabolic activity.

To further test our hypothesis, we compared TEs expression in femurs from healthy donors ($n = 27$), osteopenic ($n = 12$), and osteoporotic patients ($n = 9$) whose bone anabolism is increasingly compromised. As shown in Fig. 2A, the global expression of TEs is reduced in patients. To further assess whether TEs expression is involved in local anabolism and associated with mineral density, we divided the femur cohort according to the DXA T-score measured in the femoral neck (FN). We divided the cohort into "high BMD" (FN T-score $> -1$) and "low BMD" (FN T-score $< -1$). Globally, TEs are upregulated in femurs with high bone mineral density (Fig. 2B). In particular, more than 90% of differentially expressed TEs subfamilies are upregulated (Fig. 2C). Interestingly, almost all SINE elements are unchanged between the two groups (Fig. 2C).

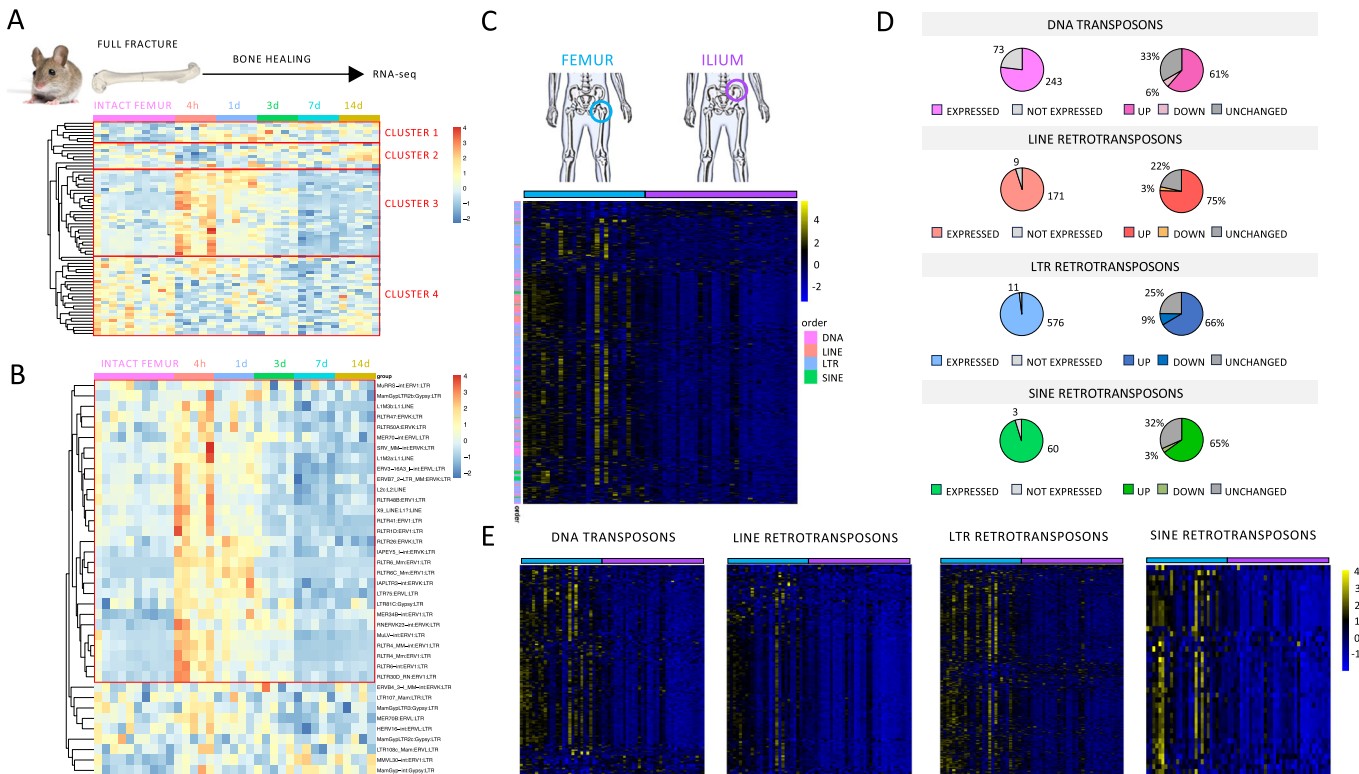

**Figure 1. TEs expression is induced after fracture and in mechanically loaded bone.**

(A) Heatmap representation of RNAseq differentially expressed TEs (LogFC >0.5) between 4 h post-injury and intact femurs at different time points of the bone healing process. N = 5 biological replicates for each time point. N = 10 biological replicates for intact femur. (B) Heatmap representation of RNAseq upregulated TEs (LogFC >0.5) between 4 h and intact at different time points of the bone healing process. N = 5 biological replicates for each time point. N = 10 biological replicates for intact femur. (C) Heatmap representation of RNAseq differentially expressed TEs analysis [fragments per kilobase of transcript per million (FPKM) fold change] in femoral (n = 27) and iliac (n = 34) bone biopsies from healthy donors. (D) Pie charts showing the number and percentage of differentially expressed TEs subfamilies between femoral and iliac healthy bone. (E) Heatmap representation of RNAseq differentially expressed TEs analysis [fragments per kilobase of transcript per million (FPKM) fold change] in femoral (n = 27) and iliac (n = 34) bone biopsies from healthy donors. One heatmap for each TEs order is shown.

Finally, we assessed whether a positive correlation exists between TEs expression levels and local mineral density (FN T-score). We found that ~30% of L1 and LTR subfamilies expressed in the femoral bone are positively correlated with local BMD (Fig. 2D). Altogether, these results strongly suggest a positive involvement of TEs in response to stress-induced anabolic demand and bone mineralization.

## Increased cytoplasmic L1 repeat RNA stimulates osteoblast mineralization activity

The in vivo evidence suggested a positive involvement of retrotransposons in posttraumatic bone repair in mice, and a strong correlation between retrotransposon expression and bone mineral density in humans, and higher expression in loaded compared to unloaded bone. We then corroborated the results by studying the effects of increasing repeat RNA levels on differentiating osteoblasts derived from mesenchymal stem cells (MSCs) isolated from femurs of healthy donors. At day 5 of differentiation, a Cy5 conjugated full-length L1 RNA consensus sequence was transfected. Capping, 2'-O-Methylation of 5' end, polyadenylation (200 adenosines), full substitution with 5-methylcytidine (m5C), and 75% substitution with pseudouridine were used to stabilize the

RNA and to bypass the intracellular innate immune system (Koski et al, 2004; Pardi et al, 2013; Ludwig et al, 2010; Karikó et al, 2005; Nallagatla and Bevilacqua, 2008; Karikó et al, 2008; Anderson et al, 2010; Kormann et al, 2011). Red fluorescent protein (RFP) mRNA, with the same modifications, was included as negative control. The exogenous L1 RNA accumulated in the cytoplasm (Fig. 3A) and was gradually cleared by secretion along with matrix components (Fig. 3B,C). The effect of L1 RNA delivery was tested in differentiating MSCs from two different and unrelated healthy donors. As shown, L1 RNA transfection markedly stimulated the production of mineralized matrix (Fig. 3D). Osteoblast mineralization increased dose-dependently with the concentration of the delivered L1 RNA, in agreement with saturation kinetics (Fig. 3E). Moreover, the strongly enhanced mineralization was highly specific for L1 RNA, as demonstrated by the lack of significant changes in mineral deposition in cells transfected with negative control RNA, even at concentrations 100–200-fold higher (Fig. 3E). We also isolated MSCs from the femur of osteoporotic patients and selected those showing a markedly delayed and reduced in vitro production of bone matrix in contrast to those derived from healthy donors (Fig. 3F). Strikingly, even in these cells with low osteogenic capacity, the delivery of L1 RNA triggered a strong mineralization response despite their clearly compromised anabolic activity

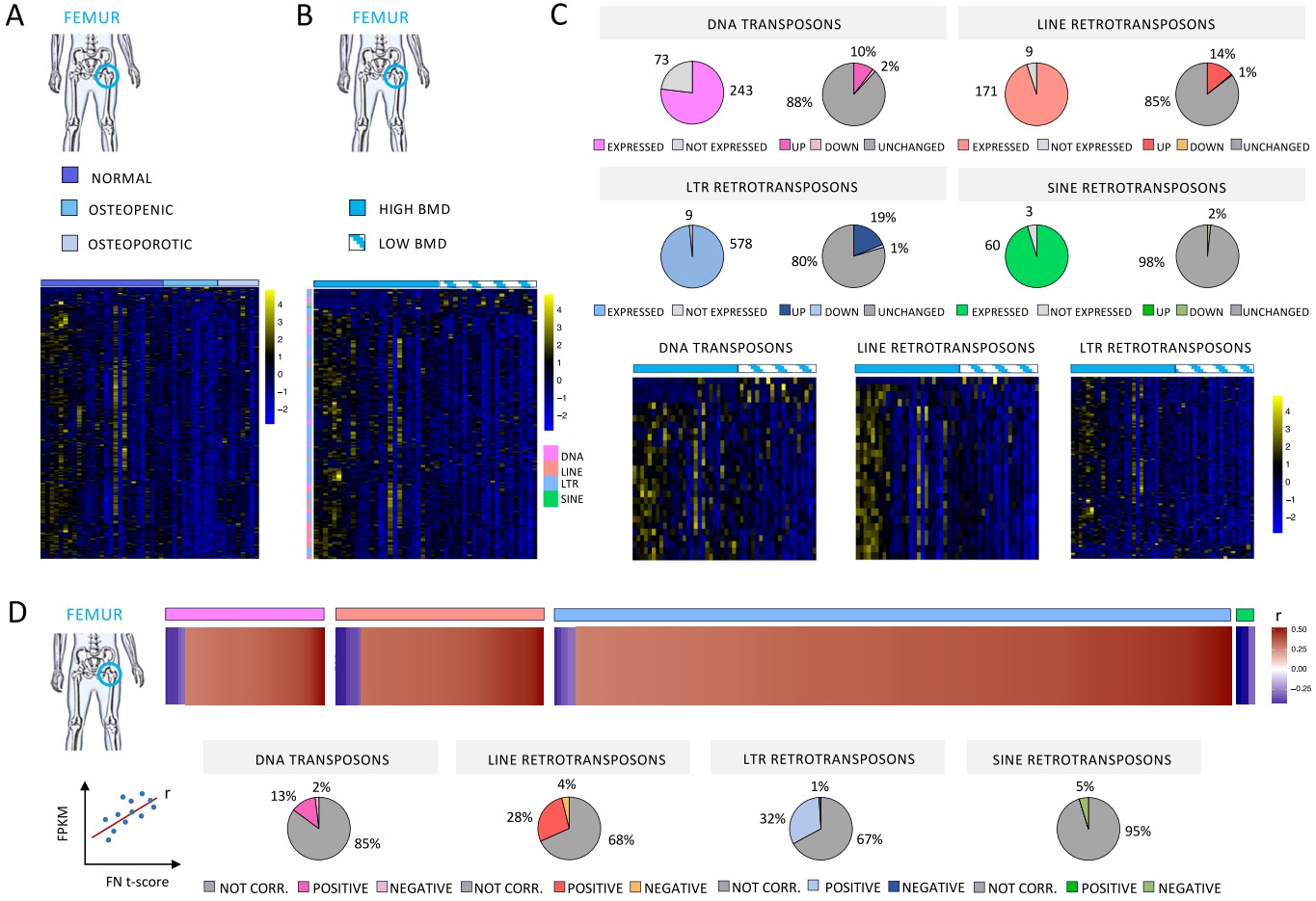

**Figure 2.  TEs expression correlates with bone mineral density in human weight-bearing bone.**

(A) Heatmap representation of RNAseq differentially expressed TEs analysis [fragments per kilobase of transcript per million (FPKM) fold change] in femoral bone from healthy (n = 27), osteopenic (n = 12), and osteoporotic (n = 9) donors. (B) Heatmap representation of RNAseq differentially expressed TEs analysis [fragments per kilobase of transcript per million (FPKM) fold change] in femoral bone with high BMD (FN T-score >−1) (n = 27) and low BMD (FN T-score <−1) (n = 21). (C) Upper panels: pie charts showing the number and percentage of differentially expressed TEs subfamilies between high BMD and low BMD femoral bone. Lower panels: heatmap representation of RNAseq differentially expressed TEs analysis [fragments per kilobase of transcript per million (FPKM) fold change] in femoral bone with high BMD (n = 27) and low BMD (n = 21). One heatmap for each TEs order is shown. (D) Upper panels: Heatmap representation of correlation analysis (P value <0.05) between TEs expression (FPKM) and local BMD (FN T-score) in femoral bone biopsies (n = 48). The correlation between TE expression (FPKM values) and FN t-score was evaluated using Pearson method. Significant correlations (p value < 0.05) were selected for the heatmap representation and arranged according to the Pearson correlation coefficient (r). r > 0, positive correlations; r < 0, negative correlations. Lower panels: Pie charts showing the percentage of TEs subfamilies positively correlated to local BMD.

(Fig. 3G). These results demonstrate convincingly that the ectopic delivery of L1 RNA stimulates the mineralization of bone-forming cells regardless of their prior intrinsic differentiation/anabolic potential.

Osteoporotic patient-derived cells transfected with L1 RNA show a higher expression of early differentiation markers, with a peak 48 h after L1 transfection (day 7) (Fig. 3H). Bone sialoprotein (*IBSP*), a late osteogenic gene, shows a similar profile, being almost 6 times more expressed in L1- than in RFP-transfected cells on day 7 (Appendix Fig. S1A). Strikingly, although alkaline phosphatase (*ALPL*) expression slightly increases at later time points (Appendix Fig. S1A), its activity is not enhanced in L1-treated cells (Appendix Fig. S2A). However, the activity of ectonucleotide pyrophosphatase/ phosphodiesterase 1 (ENPP1) is gradually and significantly reduced by L1 RNA transfection (Appendix Fig. S2B). As ENPP1 is a major source of inorganic pyrophosphate (PPi), one of the main

physiological inhibitors of mineralization (Fleisch et al, 1966; Johnson et al, 2000; Hessle et al, 2002), the reduction of its activity may contribute to the enhanced mineralization observed in L1-RNA transfected cells.

Altogether, these data suggest that L1 RNA induces a unique mineralizing phenotype and is apparently distinct from the well-known canonical differentiation mechanisms.

## L1 RNA delivery induces inflammatory pathways significantly overlapping those involved in bone fracture repair

To investigate the molecular response of osteoblasts to L1 RNA transfection, we profiled their transcriptome by Illumina RNA-Seq 24 h post-transfection. We found 482 differentially expressed genes (DEG) (FDR <0.05) between L1 and RFP RNA-treated osteoblasts

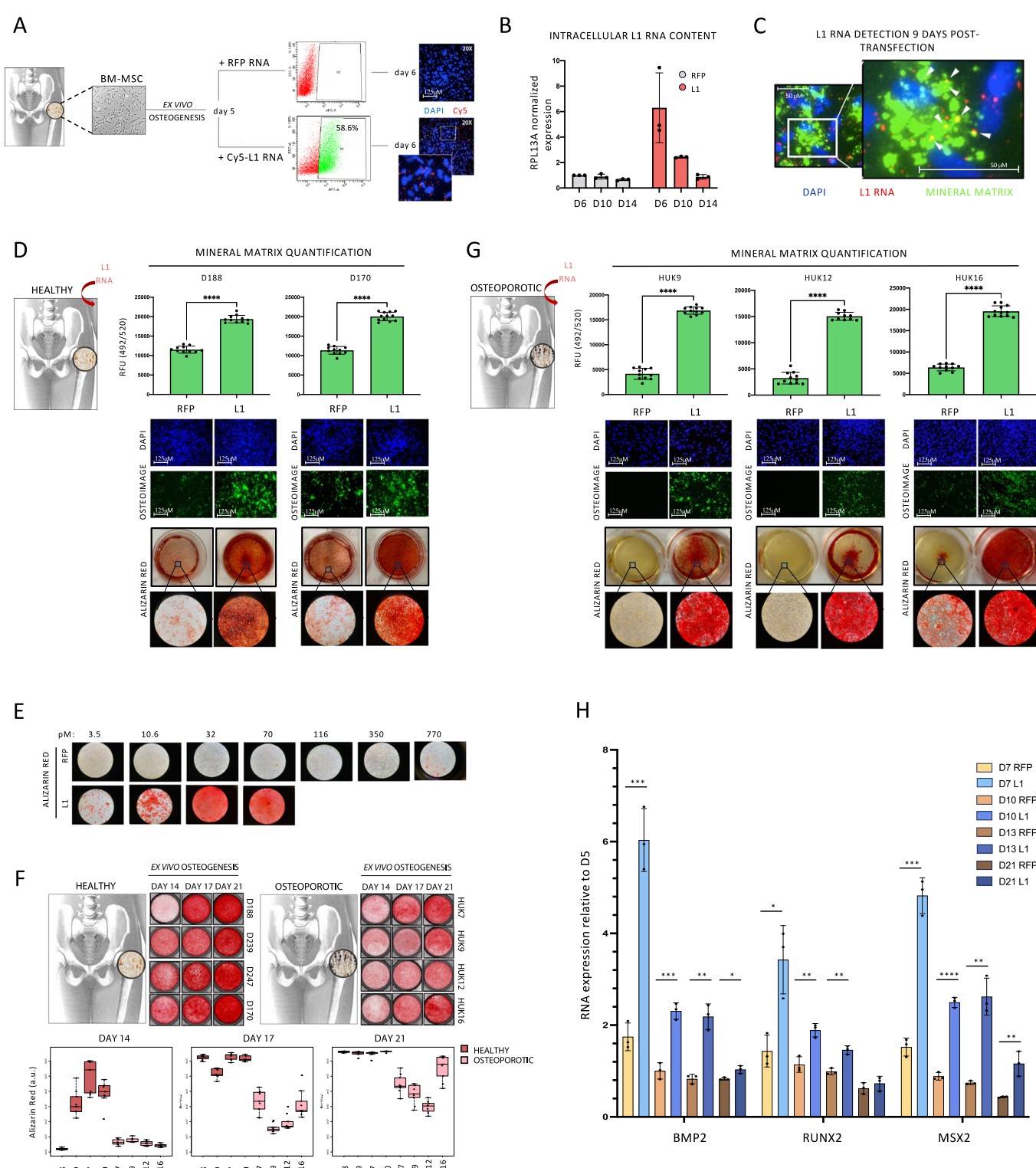

(268 upregulated and 214 downregulated) (Appendix Fig. S3). Gene ontology (GO) enrichment analysis revealed that the early transcriptional signature of L1-treated osteoblasts is typical of that of an inflammatory response (Fig. 4A,B). This is in accordance with the recent evidence of a link between L1 and other retrotransposons accumulation in the cytoplasm and inflammatory response in vitro

and in vivo (Fukuda et al, 2021; De Cecco et al, 2019b; Simon et al, 2019). GO term enrichment analysis comparison between the early stage of fracture-induced bone healing in vivo (Coates et al, 2019) and differentiating osteoblasts in vitro 24 h post-L1 delivery (Fig. 4A) revealed a matching inflammatory response and shared upregulated pathways (Fig. 4C) essential for effective bone repair

◄ **Figure 3. L1 RNA delivery stimulates the mineralization of differentiating osteoblasts.**

(A) Experimental workflow and flow cytometer analysis showing the percentage of positive cells 6 h after L1 RNA delivery at day 5 of ex vivo osteogenesis. Intracellular localization of cy5 conjugated synthetic L1 RNA (red spots) three days after transfection is also shown from a typical experiment (right). (B) qPCR analysis of intracellular L1 RNA level 24 h (d6), 5 days (d10), and 9 days (d14) post-transfection. The graph is shown as mean ± sd of $n = 3$ independent experiments. (C) cy5 conjugated synthetic L1 RNA (red) and osteoimage stained mineral matrix (green) detection 9 days post-L1 RNA transfection. The white arrowheads indicate the colocalization between L1 RNA and hydroxyapatite. (D) Osteoimage stained mineral matrix quantification (upper panels), images (central panels), and Alizarin Red (lower panels) 9 days after L1 RNA or RFP RNA delivery in two healthy donors-derived MSCs (D188 and D170). RFU relative fluorescence units. The graph is shown as mean ± sd of $n = 11$ technical replicates. ****$P < 0.00005$ in the Wilcoxon test. (E) Alizarin Red staining of osteoblasts transfected with increasing doses of RFP RNA (upper panels) and L1 RNA (lower panels). (F) Alizarin Red images (upper panel) and quantification (lower panel) of MSCs mineralization after 14, 17, and 21 days of ex vivo differentiation. MSCs were obtained from the femur of four healthy (D188, D239, D247, and D170) and four OP patients (HUK7, HUK9, HUK12, and HUK16). $N = 9$ technical replicates for each donor and time point. Boxes in the boxplot indicate the interquartile range (50% of data), while the lower end and the upper end represent the first and the third quartile, respectively. The solid line inside the box represents the median. Whiskers represent the max and min values. Values that are not within 1.5 times the interquartile range are considered outliers and lie outside the whiskers. (G) Osteoimage stained mineral matrix quantification (upper panels), images (central panels), and Alizarin Red (lower panels) 9 days after L1 RNA or RFP RNA delivery in three OP patients derived MSCs (HUK9, HUK12, and HUK16). RFU relative fluorescence units. The graph is shown as mean ± sd of $n = 10$–12 technical replicates. ****$P < 0.00005$ in the Wilcoxon test. (H) qRT-PCR of early osteogenic genes in RFP and L1-transfected osteoblasts at different time points of osteogenic differentiation. Expression level is normalized on day 5 (not transfected osteoblasts). $N = 3$ biological replicates. The graph is shown as mean ± sd of $n = 3$ independent experiments. *$P < 0.05$; **$P < 0.005$, ***$P < 0.0005$, ****$P < 0.00005$ in Student's $t$-test. Source data are available online for this figure.

(Maruyama et al, 2020; Gerstenfeld et al, 2003; Kon et al, 2001). Recruitment of neutrophils at the site of injury also indicates a healthy repair process, particularly in bone (Kovtun et al, 2016). Extracellular signal-regulated kinase 1 (ERK1) and 2 (ERK2), play crucial roles in bone formation (Xiao et al, 2000; Xiao et al, 2002b; Xiao et al, 2002a; Matsushita et al, 2009; Ge et al, 2007) and seem to be the most upstream initiators of tissue regeneration in planaria (Owlarn et al, 2017). Upon bone injury, an initial transient stage of acute inflammation is a crucial factor in ensuring effective regeneration (Maruyama et al, 2020), whereas an excessive/prolonged (chronic) inflammation is deleterious for the healing environment (Chan et al, 2012; Hurtgen et al, 2016; Clark et al, 2017; Brem and Tomic-Canic, 2007; Timmen et al, 2014; Schmidt-Bleek et al, 2012; Weckbach et al, 2013). We followed the time-course expression of "inflammatory response" (GO:0006954) and "immune response" (GO:0006955) genes upregulated by L1 RNA immediately after transfection. As shown in Fig. 4D, their expression is already strongly reduced at day 10 and almost entirely silenced by day 14, suggesting the transient dynamics of the inflammatory response triggered in vitro by L1 RNA. This is coherent with our previous observation that exogenous L1 RNA is gradually secreted by the cell together with matrix components (Fig. 3B,C), eventually clearing out the initial inflammatory stimulus.

## Cellular response to increased cytoplasmic level of L1 RNA is mediated by PKR

It has been previously demonstrated that the cytoplasmic accumulation of L1-derived cDNA triggers an inflammatory response mediated by the cyclic GMP-AMP synthase (cGAS)-stimulator of interferon genes (STING) cytosolic DNA sensing pathway (De Cecco et al, 2019a). Immunofluorescence (IF) analysis reveals that the exogenous L1 RNA forms DNA:RNA hybrids (Appendix Fig. S4), suggesting that L1 RNA is reversed transcribed into cDNA after its transfection. Notably, ORF1p was undetectable by WB in differentiating osteoblasts and in L1-transfected osteoblasts (Appendix Fig. S5), indicating that the exogenous L1 RNA is not translated after transfection. To assess whether L1-induced inflammation was triggered by the sensing of its cDNA, we used Lamivudine 3TC and G140 (Fig. 5A) to inhibit the ORF2p-mediated reverse transcription of L1 RNA and

cGAS activity, respectively. Moreover, we also performed a siRNA-mediated knockdown (KD) of cGAS (Appendix Fig. S6) (Jones et al, 2008; Lama et al, 2019). Surprisingly, none of the treatments impaired the mineralization induced by L1 RNA delivery (Fig. 5A; Appendix Fig. S6C), suggesting that the observed cellular response depends entirely on RNA sensing. As previously mentioned, repeats are a major source of endogenous dsRNA (Sadeq et al, 2021; Chen and Hur, 2022), and L1 RNA, in particular, is able to form intramolecular double-stranded structures (Hur, 2019). As expected, we found a tight colocalization between L1 RNA and dsRNA IF signal (Fig. 5B). One of the main cellular sensors of dsRNA, including those derived from retrotransposons, is PKR (Kim et al, 2018). PKR activation results in the phosphorylation of eIF2α and in the subsequent inhibition of global protein synthesis and cell growth (Donnelly et al, 2013). Interestingly, PKR has been shown to interact with L1 RNA in vitro (Kim et al, 2018). Moreover, eIF2α phosphorylation has been demonstrated to promote autophagy in osteoblasts and to counteract BMD reduction in osteoporotic ovariectomized mice (Li et al, 2019).

To demonstrate that L1 RNA is able to trigger a PKR-mediated intracellular response, we transfected L1 RNA in the presence or absence of the PKR inhibitor C16 and evaluated the levels of eIF2α phosphorylation by western blot (Fig. 5C). The delivery of L1, but not RFP, strongly induces eIF2α phosphorylation and this effect is prevented by PKR inhibition (Fig. 5C). Notably, eIF2α phosphorylation occurs already 6 h post-transfection. This observation suggests that PKR activation is one of the first events induced by L1 RNA, thus corroborating the hypothesis of PKR as an endogenous L1 RNA sensor.

We then characterized the proteome of cells 24 h post-L1 RNA transfection by diaPASEF mass spectrometry (MS) (Meier et al, 2020a). We detected almost 8000 proteins in each sample, and observed that more than 88% of significantly differentially expressed proteins (DEP) were downregulated in L1-treated cells (Fig. 5D, left panel). Moreover, among the top 15 downregulated biological processes and cellular components were translation and ribosomal subunits (Fig. 5D, right panels, yellow circles), indicating that L1 RNA induces a global attenuation of protein synthesis. Indeed, a cap-dependent translation shutdown is the main consequence of eIF2α phosphorylation (Donnelly et al, 2013). Finally, we show that both PKR KD (Appendix Fig. S6A,B) and its pharmacological inhibition by C16

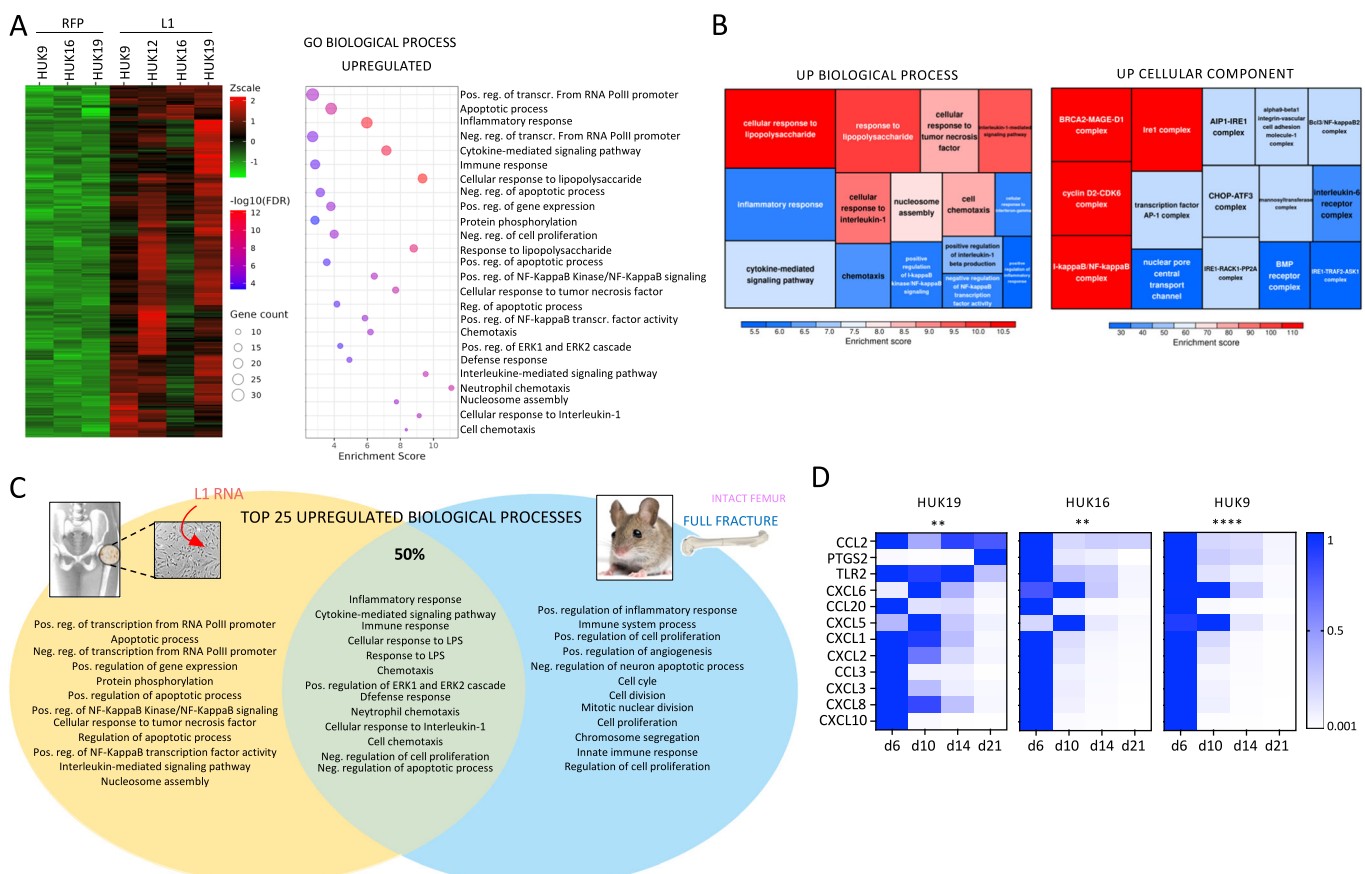

**Figure 4. L1 RNA delivery induces an inflammatory response characteristic of the bone repair process.**

(A) Left panel: heatmap representation of RNAseq differentially expressed gene analysis [fragments per kilobase of transcript per million (FPKM) fold change] in osteoblasts transfected with negative control RNA (RFP) or L1 RNA. Right panel: bubble plot showing gene ontology (GO) enrichment analysis of L1 upregulated biological processes. (B) Tree plot showing gene ontology (GO) enrichment analysis of L1 upregulated biological processes (left) and cellular components (right). (C) Top 25 upregulated biological processes 24 h post-L1 RNA delivery in vitro (yellow) and 4 h post fracture in vivo (blue). Shared GO terms are shown in green. (D) qPCR analysis of "inflammatory response" (GO:0006954) and "immune response" (GO:0006955) genes at different time points post-L1 RNA transfection at day 5. ***$P < 0.005$, ****$P < 0.00005$ in two way ANOVA. Source data are available online for this figure.

treatment (Fig. 5E,F) prevents L1 RNA-triggered induction of inflammatory genes (Fig. 5E) and mineral matrix deposition (Fig. 5F; Appendix Fig. S6B). As previously mentioned, a global reduction of protein synthesis occurs 24 h after L1 transfection, and only a little percentage of DEP is upregulated in L1 compared to RFP control (Fig. 5D). Interestingly, some of the most upregulated proteins are involved in autophagy and vesicle trafficking (Fig. 5G). Autophagy is a crucial process for mineralization and bone homeostasis both in vitro and in vivo (Nollet et al, 2014), as autophagic vacuoles are exploited to secrete apatite crystals.

Altogether these data demonstrate that the PKR-mediated sensing of L1-derived dsRNA globally attenuates the translation via eIF2α phosphorylation and indicate autophagy as a possible mechanism involved in L1-induced osteoblasts mineralization.

## L1 RNA delivery reshapes the secretory profile of differentiating osteoblasts

An inflammatory response translates into a strong secretory activity and production of paracrine signals, and we, therefore, tested the

effect of "L1-primed" osteoblast-derived conditioned medium in recipient differentiating osteoblasts (Fig. 6A). We found that the "secretome" of cells transfected with L1 RNA has a significant paracrine effect on recipient osteoblasts since they showed an earlier formation of mineralized nodules, already 24 h post the addition of conditioned media (Fig. 6A, right panel), and higher deposition of a mineral matrix at day 10 (Fig. 6A, left panel). We subsequently isolated bulk and exosome-derived proteomes from conditioned media of untreated, RFP- and L1-treated differentiating osteoblasts and characterized them by MS. Differential expression analysis of MS data revealed a unique secretome profile (for both bulk and vesicular proteomes) of L1-primed osteoblasts compared to RFP-primed and untreated osteoblasts (Fig. 6B). Gene Ontology (GO) analysis of differentially expressed proteins showed enrichment of proinflammatory factors (i.e., interleukins and chemokines) involved in immune response and chemotactic migration of immune cells (Fig. 6C,D), crucial processes for tissue repair mechanisms in vivo. Proinflammatory molecules are also a major constituent of the senescence-associated secretory phenotype (SASP) (Coppé et al, 2010), whose transient delivery supports

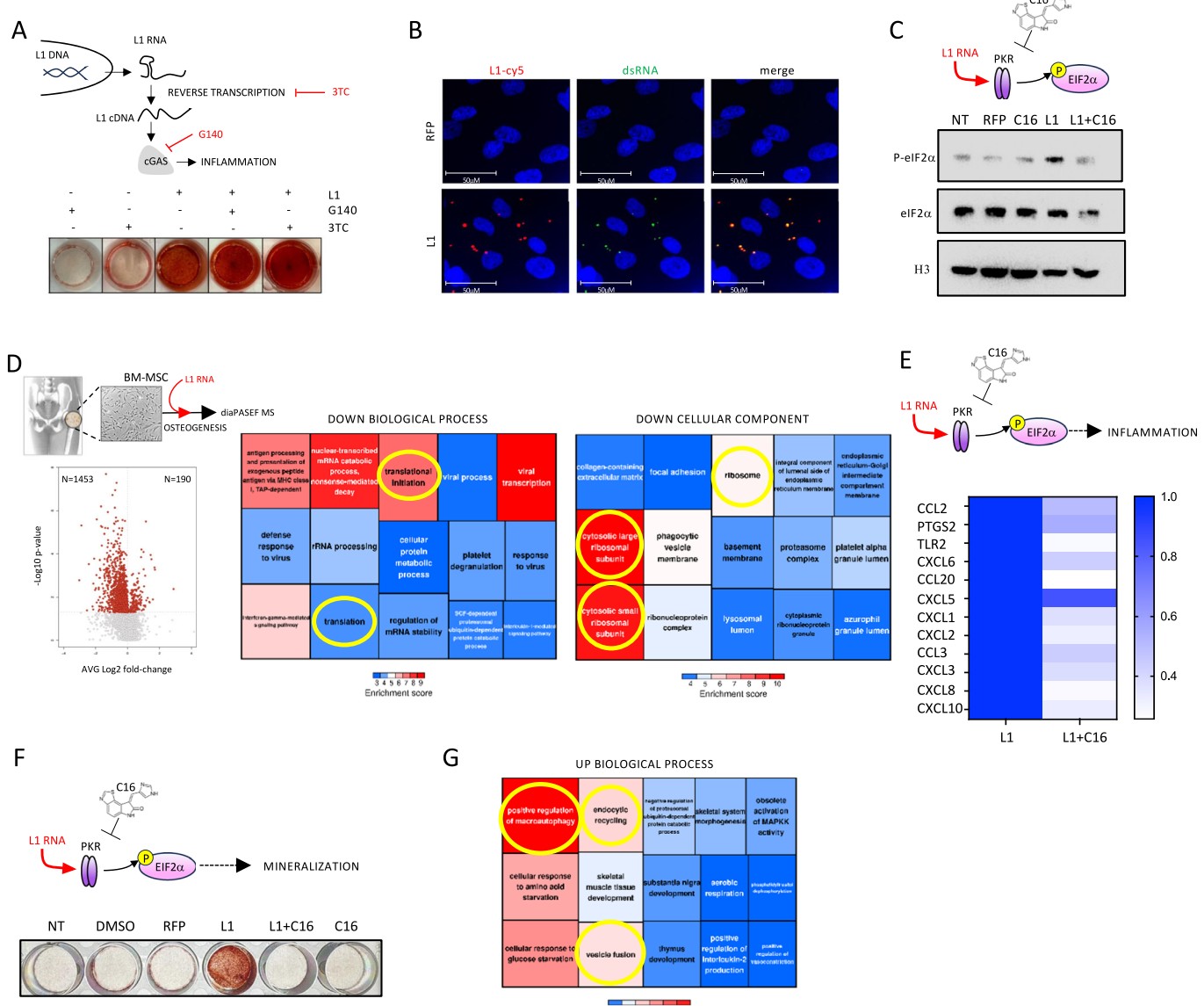

**Figure 5. PKR mediates L1 RNA-induced stress response and mineralization.**

(**A**) Upper panel: schematic representation of the pathway inhibited by Lamivudine 3TC and G140. Lower panel: Alizarin Red staining of differentiating osteoblasts transfected with L1 RNA and treated with Lamivudine 3TC and G140. (**B**) IF on RFP and L1-transfected osteoblasts, 6 h post-transfection, shows the colocalization between the cy5 signal (L1 RNA, red) and 488-Anti-dsRNA signal (dsRNA, green). Nuclei are stained with Hoechst (blue). (**C**) Western blot analysis showing the ratio between total eIF2α and phosphorylated eIF2α (P- eIF2α). H3: endogenous standard Histone 3. (**D**) Volcano plot (left) showing the number of significantly upregulated and downregulated protein 24 h post-L1 RNA transfection. Right: tree plots showing gene ontology (GO) enrichment analysis of biological processes (upper panel) and cellular components (lower panel) downregulated by L1 RNA delivery (MS data). (**E**) qPCR analysis of "inflammatory response" (GO:0006954) and "immune response" (GO:0006955) genes 24 h post-L1 RNA transfection with and without PKR inhibitor C16. (**F**) Alizarin Red staining of differentiating osteoblasts transfected with L1 RNA with and without PKR inhibitor C16. (**G**) Tree plots showing gene ontology (GO) enrichment analysis of biological processes upregulated by L1 RNA 24 h after transfection (MS data). Source data are available online for this figure.

cellular plasticity and tissue regeneration (Ritschka et al, 2017). Notably, the most enriched protein in L1-specific bulk secretome is interleukin 8 (IL-8), an inflammatory chemokine involved in several regenerative processes, such as skin wound healing (JIANG et al, 2012), therapeutic angiogenesis after stroke and ischemia (Choi et al, 2021; Hou et al, 2014) and osteochondral bone repair (Lin et al, 2019; Yang et al, 2018a). Tumor necrosis factor-inducible gene 6 (TSG6) is among the top ten proteins enriched in the bulk

secretome of L1-primed osteoblasts compared to RFP. TSG6 is an inflammatory factor with suggestive therapeutic effects in corneal wounds, myocardial infarction, injured central nervous system, chronic liver damage, and intervertebral disc degeneration (Zhang et al, 2013; Lee et al, 2015; Lee et al, 2009; Wang et al, 2017; Yang et al, 2018b). Moreover, TSG6 induces autophagy influx both in vivo and in vitro (Wang et al, 2017). Another protein positively involved in autophagy and found specifically in the secretome

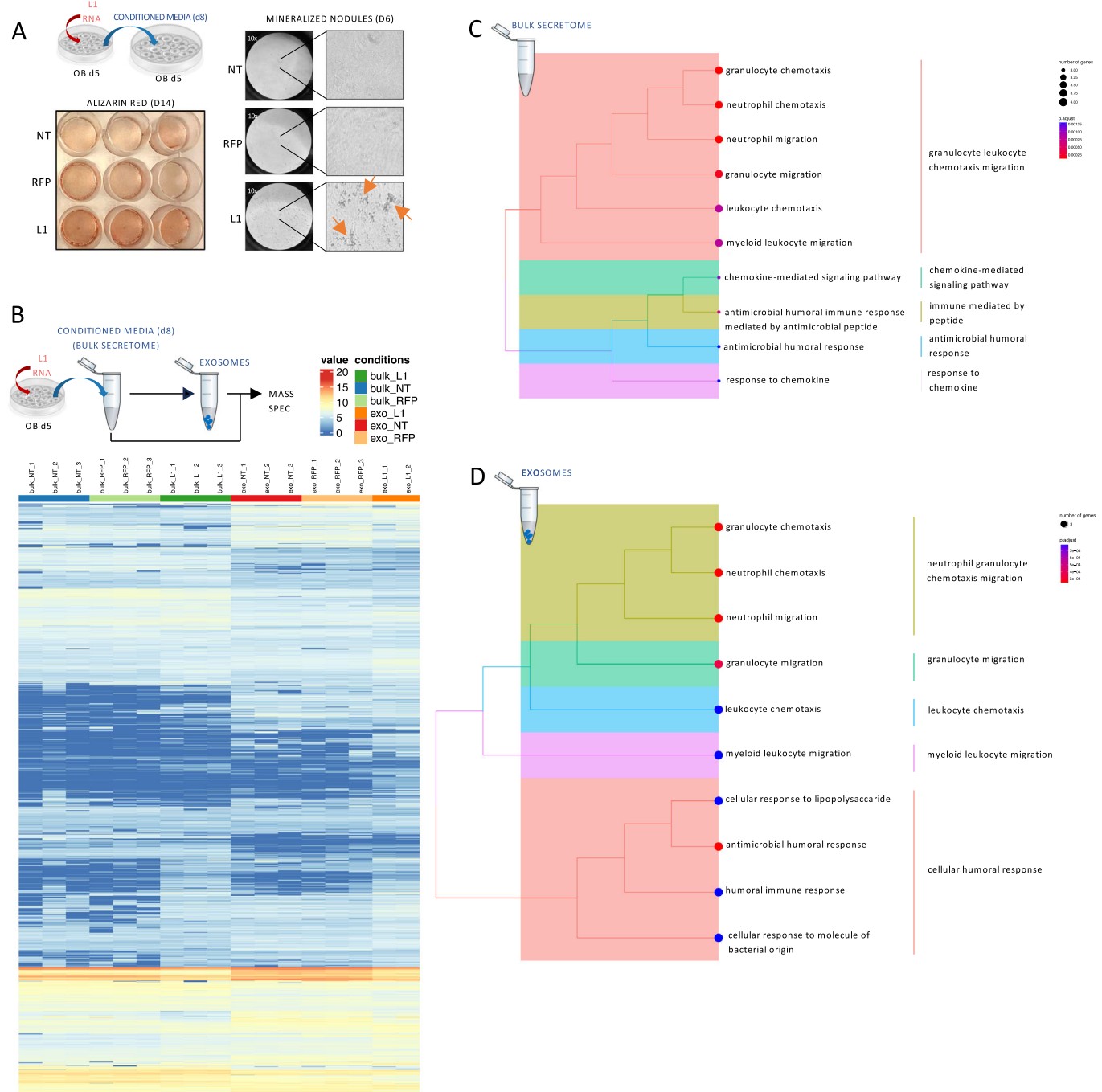

**Figure 6. L1-RNA-induced changes in osteoblast secretome.**

(A) Left: Alizarin red staining on recipient osteoblasts (OB) 9 days after the delivery of conditioned media. Right: Microscope images of recipient OB 24 h from the delivery of conditioned media. Mineralized nodules in OB receiving conditioned media from L1-treated OB are shown (orange arrows). (B) Heatmap of differentially expressed proteins in bulk secretome and exosomes derived from untransfected (NT), RFP- and L1-transfected osteoblasts. N = 3 biological replicates. (C) GO enrichment analysis of differentially expressed protein (adjusted *p* value <0.05) in the bulk secretome of L1 compared to RFP-transfected osteoblasts. (D) GO enrichment analysis of differentially expressed protein (adjusted *p* value <0.05) in the exosomes of L1 compared to RFP-transfected osteoblasts. Source data are available online for this figure.

of L1-primed osteoblasts was ATG7 (autophagy-related 7). Osteoblast-specific ATG7 conditional knockout in mice reduced bone mass during developmental and adult age (Li et al, 2018). Type 1 lysophosphatidic acid receptor (LPAR1) is also specifically secreted by L1-primed osteoblasts and has proven to be positively involved in bone mineralization in vitro and in vivo (Alioli et al, 2020; Gennero et al, 2011).

Altogether, our findings demonstrate that cytoplasmic L1 RNA is sensed by PKR, whose activation leads to eIF2α phosphorylation with consequent reprogramming of transcription and translation.

The ultimate consequence of this L1-induced, PKR-mediated stress response in osteoblasts is a major reprogramming of their secretory activity, leading to a paracrine effect on recipient osteogenic competent cells to trigger their mineralization activity.

# Discussion

The concerted co-option of TEs markedly changed whole regulatory networks and integrated new functions into the eukaryote genome (Cosby et al, 2019; Mangiavacchi et al, 2021; Carelli et al, 2022). However, while the role of TEs as evolutionary drivers is today well-established, their contribution to cell physiology, particularly in somatic cells, remains to be elucidated. Indeed, aside from their role in the nucleus (Della Valle et al, 2022; Jachowicz et al, 2017; Lu et al, 2021; Percharde et al, 2018), TEs represent a major source of endogenous dsRNA whose essential role in inflammatory and innate immune responses involved in various physiological processes has increasingly gained attention (Sadeq et al, 2021; Chen and Hur, 2022; Angileri et al. 2022). For instance, an early acute inflammatory response is of vital importance in initiating and orchestrating bone repair and induce anabolism leading to fracture healing (Mountziaris et al, 2011; Marsell and Einhorn, 2011; Bahney et al, 2019). Moreover, several studies demonstrated a positive effect of proinflammatory mediators on bone mineralization in vitro and in vivo (Croes et al, 2015; Croes et al, 2016; Li et al, 2016; Croes et al, 2017), indicating that the modulation of inflammatory reaction is a tool to pursue regenerative strategies in bone (Roberts and Ke, 2018). We found that, in the bone of newly fractured mice, a group of TEs, mostly LINE and LTR, was early, markedly and transiently upregulated. Interestingly, the expression of these TEs was limited to the earliest phase of the inflammatory stage. In line with the hypothesis of a positive correlation between stress-induced bone mineralization and TEs activation, the expression dynamics in human bone biopsies revealed a significant upregulation of TEs in the femur, a bone subjected to high mechanical stresses, high turnover and anabolism, compared to ilium, not weight bearing. Furthermore, we detected a lower TEs expression in donors with compromised bone mineralization and a significant correlation between TEs expression and local BMD. We hypothesized that the expression dynamics of TEs in fracture/mechanical stress observed in vivo could mimic the immunological threshold model of the sterile activation of dsRNA sensors as a response to stressful conditions. Indeed, the transient and controlled breaching of the activation threshold of dsRNA sensors may lead to sterile inflammation being integrated into physiological processes (Chen and Hur, 2022). To test further this hypothesis, we delivered a full-length L1 consensus sequence as source of dsRNA (Hur, 2019) to differentiating osteoblasts in vitro and showed that L1 RNA is initially accumulated in the cytoplasm and then it is gradually cleared by secretion. The increased cytoplasmic levels of L1 RNA are immediately sensed by PKR, a cytoplasmic dsRNA sensor whose activation induces the phosphorylation of eIF2α (Kim et al, 2018) (Donnelly et al, 2013), leading to the attenuation of global protein synthesis (Donnelly et al, 2013), induction of inflammatory response, and stimulation of mineral matrix deposition. In support of the evidence that PKR is the early sensor of increased L1 RNA cytoplasmic levels, we show that inhibition of PKR activation by C16 prevents L1-induced eIF2α

phosphorylation, induction of inflammatory genes, and mineral matrix deposition. Moreover, the inhibition of L1 cDNA formation and sensing by Lamivudine 3TC, G140, or cGAS KD does not affect the L1-induced phenotype. Thus, in our system, the evidence does not support that the L1-triggered inflammation is causally connected to cGAS-STING-dependent sensing of cytosolic L1 cDNA, a mechanism demonstrated operative in senescence (De Cecco et al, 2019a).

L1 RNA response showed high specificity, as demonstrated by the dose-dependent increase of mineralization and the lack of significant changes in mineral deposition when negative control RNA was transfected, even at 100–200-fold higher concentration. Although the molecular mechanism behind L1-induced stress response which stimulates the mineralization activity of osteoblasts remains elusive, our results indicate that L1 stimulates the mineral matrix deposition by bone-forming cells regardless of their prior intrinsic anabolic potential. Moreover, MS results suggested that increased autophagy, a process crucial for mineralization and bone homeostasis in vitro and in vivo (Nollet et al, 2014) may contribute to L1-induced hydroxyapatite deposition. Intriguingly, the osteoinductive properties of L1-primed osteoblast-derived secretome and the observation that L1 RNA was gradually secreted by the cells may suggest an involvement of L1 RNA in cell-to-cell communication, in line with the recent finding of other retrotransposons (i.e., ERV) acting as paracrine molecules (Liu et al, 2021). This, together with the observed L1-dependent induction of CCL and CXCL chemokines and cytokines, which are chemoattractants of immune cells and endothelial cells (Bischoff et al, 2011; Eash et al, 2010; Strieter et al, 2005), and the composition of L1-primed osteoblast secretome, makes the in vivo delivery of L1 RNA a necessary step to evaluate its actual contribution to the paracrine processes that orchestrate bone repair mechanisms (Chaparro and Linero, 2016; Nimiritsky et al, 2019).

The identification of repeats RNA, particularly L1, as potential resilient molecular factors involved in stress-induced, inflammation-mediated bone production points out new avenues to develop anabolic strategies for the treatment of bone loss conditions, such as osteoporosis or impaired bone repair (Roberts and Ke, 2018). On the other hand, repeats may be novel targets in the treatment of those conditions characterized by excessive or heterotopic ossification preceded by deregulated/chronic inflammation (Lories and Schett, 2012; Balboni et al, 2006).

# Methods

## Participants and ethics

Femoral bone biopsies were obtained from the caput region of postmenopausal women or men with a wide BMD range, i.e., from healthy to osteoporotic, who were undergoing hip replacement surgery due to osteoarthritis or fracture at Lovisenberg Diaconal Hospital (Oslo, Norway) or Diakonhjemmet Hospital (Oslo, Norway), respectively. The donors of femoral bones are listed in Appendix Table S1. The postmenopausal iliac bone donors were recruited from the outpatient clinic of Lovisenberg Diaconal Hospital (Oslo, Norway). Candidates filled out a questionnaire that included medication and lifestyle factors, and selected donors were deemed representative of the Oslo-based Norwegian ethnic

female population aged 50 to 86 years. The iliac bone donors are listed in Appendix Table S2 and have been described previously in detail (Reppe et al, 2010). All donors taking medication or having diseases, other than primary osteoporosis, known to affect bone metabolism were excluded. The presence of bone-impairing diseases/conditions was excluded by extensive biochemical serum and urine analyses supported by X-ray examinations. The site-specific BMD of all donors was evaluated with Lunar Prodigy DEXA (GE Lunar, Madison, WI, USA) following the manufacturer's instructions. The precision of the instrument for measuring the lumbar spine ($L_2$–$L_4$) and hip BMD was 1.7 and 1.1%, respectively. The study was approved by the Norwegian Regional Ethical Committee (REK no 2010/2539, Norway), all volunteers gave their written informed consent, and sampling and procedures were according to the Act of Biobanking in Norway.

## Cell culture

For L1 RNA transfection experiments, human MSCs were isolated from osteoporotic femoral heads with the consent of the patient according to Swiss (BASEC-Nr. 2016-02159) ethical guidelines. Femoral heads were crushed and incubated in DMEM (PAN Biotech, Germany, Cat. No. P04-03550) supplemented with 1% (v/v) 100× Penicillin-Streptomycin Solution (Biowest, France, Cat. No. L0022), 10% (v/v) FCS (Biowest, France, Cat. No. S181S), 1% (v/v) 200 mM L-glutamine solution (Sigma, USA, Cat. No. G7513), 5 ng/ml FGF-2 (Sigma, USA, Cat. No. F0291), and 10 ng/ml FGF-4 (Sigma, USA, Cat. No. F8424) in a humidified atmosphere containing 5% $CO_2$. To induce osteogenic differentiation, MSCs were seeded on Nunc™ 24-well plates (Thermo Fisher, USA, Cat. No. 142475) or 48-well plates (Thermo Fisher, USA, Cat. No. 150687) at a density of $1.5 \times 10^4$ cells/cm². After 24 h, differentiation was induced using StemPro® Osteogenesis Kit (Gibco/Life Technologies, USA, Cat. No. A10072-01). The medium was exchanged every 3 days. Cells were treated with Lamivudine 3TC (Sigma), 1 μM final concentration; C16 (Merck), 500 nM final concentration; G140 (InvivoGen), and 150 μM final concentration.

## L1 RNA transfection

The vector human-L1_pBluescript II sk (+) carrying the full-length L1 sequence was custom-prepared by GenScript, USA. Large-scale human L1 mRNA was in vitro transcribed, modified, and purified by TriLink Biotechnologies, USA, (ARCA capped and 2'Omethyl-malted (CapI), fully substituted with 5-methyl-C, 25% substitution of Cyanine-5-U and 75% substitution of Pseudo-U, enzymatically polyadenylated, DNase and phosphatase treated, silica membrane purified). Synthetic L1 RNA was transfected in MSCs differentiating to osteoblasts at day 5 using Lipofectamine™ MessengerMAX™ (Invitrogen, USA, Cat. No. LMRNA003) with a modified protocol for low RNA amount. RFP mRNA (System Bioscience, USA, Cat. No. MR800A-1) was used as a negative control. Bone matrix was quantified with OsteoImage Mineralization Assay (Lonza, Basel, Switzerland, Cat. No. LOPA503) and Alizarin Red staining. In conditioned media experiments, the medium was exchanged 6 h after transfection. Conditioned medium was collected after three days and delivered to the recipient, differentiating osteoblasts at day 5. In knockdown experiments, siRNAs were transfected with Lipofectamine RNAiMAX (Thermo Scientific) in differentiating osteoblasts at day 3, followed by L1 RNA transfection at day 7.

siRNAs: EIF2AK2- Targeting SMARTpool (Dharmacon, Horizon, E-003527-00-0010), MB21D1-Targeting SMARTpool (Dharmacon, Horizon, L-015607-02-0020), Non-targeting Pool (D-001810-10).

## Quantitative mineralization assay

Cells were washed in PBS and fixed in 4% (v/v) formaldehyde (Sigma, USA, Cat. No. F8775) in 1× PBS for 15 min. Mineralization was assessed by using the OsteoImage Mineralization Assay (Lonza, Basel, Switzerland, Cat. No. LOPA503) according to the manufacturer's indication. Mineralization was quantified with GloMax® discover microplate reader (Promega, USA), selecting appropriate excitation (492)/emission (520) wavelengths.

## Alizarin Red staining

Osteoblasts were washed with 1× PBS (Kantonsapotheke Zürich, Switzerland, Cat. No. A171012) and fixed with 4% (v/v) formaldehyde (Sigma, USA, Cat. No. F8775) in 1× PBS for 30 min. After washing twice with ddH2O, Alizarin Red staining solution (0.7 g Alizarin Red S (Sigma, USA, Cat. No. A5533) diluted in 50 ml ddH$_2$O at pH = 4.2) was added for 20 min. Afterward, cells were washed four times with ddH2O, dried, and stored in the dark until image acquisition. For absorbance measurement, Alizarin Red S was eluted from stained osteoblasts with 300 μl 10% (w/v) cetylpyridinium chloride in an aqueous 0.01 M $Na_2HPO_4$/$NaH_2PO_4$ solution at pH = 7 for 1 h. One hundred fifty microliters were transferred on a 96-well plate, and absorbance was measured at 560 nm. Ten percent (w/v) cetylpyridinium chloride in an aqueous 0.01 M Na$_2$HPO4/NaH2PO4 solution was used as blank. Images were acquired, processed, and analysed as previously described (Eggerschwiler et al, 2019).

## ALPL activity assay

ALPL activity was measured as in Liu et al, 2016. Briefly, cells were fixed in 3.7% formaldehyde at RT for 10' and then stained with a solution of 25% naphthol AS-BI phosphate (Thermo) and 0.75% Fast Blue BB (Sigma) dissolved in 0.1 M Tris buffer (pH 9.3) at RT for 15'. After staining, cells were quickly washed with PBS five times.

## ENPP1 activity assay

ENPP1 activity was calculated colorimetrically using the chromogenic substrate p-nitrophenyl-thymidine-5'-monophosphate, as described in (Ferreira et al, 2013). After PBS washing, cells were lysed in a solution of 0.1% Triton X-100, 0.2 M Tris-base, 1.6 mM $MgCl_2$, pH 8.1. About 50 μl of 1 mM thymidine monophosphate p-nitrophenyl ester (Sigma) were added to 50 μl of cell lysate and incubated for 1 h at 37 °C. Then, four volumes of 0.1 M NaOH were added to stop the reaction and absorbance was read at 410 nm using a spectrophotometer (Tecan Nanoquant Infinite 200 Pro Multimode). Absorbance was normalized to DNA content.

## Immunohistochemistry

Cells were fixed in 4% paraformaldehyde in PBS at room temperature (RT) for 10'. After fixation, cells were permeabilized in 3% Triton X-100 in PBS for 3' at RT. Blocking was performed

with 4% BSA in PBS for 30' at RT. After blocking, cells were incubated with primary antibodies overnight at 4 °C, then washed with PBS and incubated with secondary antibodies for 1 h at RT. After washing with PBS, nuclei were stained with Hoechst (Thermo Scientific) and mounted using Fluoro Gel with DABCO™ Mounting Medium (Electron Microscopy Sciences). Primary antibodies: DNA-RNA Hybrid (mouse, Merck Millipore, MABE1095), dsRNA (mouse, Merck Millipore, MABE1134).

## RNA extraction and cDNA preparation

Cells were harvested and resuspended in 1 ml of QIAzol Lysis reagent (Qiagen, Cat. No. 79306). Total RNA was then purified with the RNeasy Plus Mini kit (Qiagen, cat. No. 74134) with minimal modifications to the manufacturer's instructions. DNase treatment (RNase-free DNase set, Qiagen, Cat. No. 79254) was performed to remove any residual DNA. RNA quality and concentration were checked using a Nanodrop™ 2000 spectrophotometer (Thermo Fisher). cDNA was synthesized from 200 ng of each RNA sample using a Superscript III first-strand cDNA synthesis system (Thermo Fisher, cat. No. 18080051) according to the manufacturer's protocol. Processing of human bone biopsies and RNA isolation has been described previously (Reppe et al, 2010).

## Gene expression analysis

Real-time quantitative polymerase chain reaction (qPCR) was performed with a 7900HT Fast Real-Time PCR system (Applied Biosystems). Each sample was analyzed in triplicate and normalized with the endogenous control Ribosomal Protein L13A (*RPL13A*) for cDNA input concentration. No template and no RT were included as negative controls. For each 15 μl reaction, 10 ng (1 ng for L1) of cDNA was mixed with 1 μM specific primers mix and 7.5 μl of Sybr™ Select Master mix (Applied Biosystems, USA, Cat. No. 4472908). The reaction was incubated at 95 °C for 10 min, followed by 40 cycles of denaturation at 95 °C for 15 s, annealing at 60 °C for 30 s, and elongation at 72 °C for 30 s. Ct values were calculated by 7900HT Fast Real-Time PCR RQ manager software (Applied Biosystems, USA) and then normalized as ΔCt between the gene of interest and the endogenous calibrator. Primers used in this study for gene expression analysis were designed using Primer3 (http://www.ncbi.nlm.nih.gov/tools/primer-blast/). In all primer pairs, each primer matches a different exon. Amplicons length was 80–130 nucleotides. Primer sequences are reported in Appendix Table S3.

## Cell cycle analysis

About $2 \times 10^5$ MSCs were trypsinized for 5 min at 37 °C, washed with PBS and 2% BSA, passed through a 70 μM strainer (Corning, USA, Cat. No. 352350), and then fixed at −20 °C for 30 min in 70% ethanol. After washing with PBS and 4% BSA, cells were resuspended in PBS and incubated for 1 h at 37 °C with RNase. Cells were then washed and resuspended in 100 μl of Flow Cytometry Staining Buffer (R&D System, USA, Cat. No. FC001). About 10 μl of 1 mg/ml Propidium iodide (PI) staining solution (Invitrogen, USA, Cat. No. P3566) was added to the single-cell solution, gently mixed, and incubated for 5 min in the dark. Cell cycle analysis was performed on BD FACSCanto II Flow Cytometry System (BD-Biosciences), using BD FACSDiva Software (BD-Biosciences).

## RNAseq and data analysis

RNA from human bone biopsies was sequenced at the human genotyping facility (HuGe-F) of Erasmus MC.

Total RNA-Seq library, from L1 RNA delivery experiments, was prepared with CORALL Total RNA library prep with RiboCop rRNA for Human/Mouse/Rat depletion kit (Lexogen GmbH, Vienna, Austria) following manufacturer's instructions (library type: fr-secondstrand) by IGA Technology service (Italy). The final libraries were checked using both Qubit 2.0 Fluorometer (Invitrogen, Carlsbad, CA) and Agilent Bioanalyzer DNA assay or Caliper (PerkinElmer, Waltham, MA). Libraries were then prepared for sequencing and sequenced on paired-end 150 bp mode on NovaSeq6000 (Illumina, San Diego, CA). RNA-Seq read quality control (QC) analyses and filtering of high-quality reads were executed using FastQC v0.11.9 (http://www.bioinformatics.babraham.ac.uk/projects/fastqc/) and BBDuk v35.85 (https://jgi.doe.gov/data-and-tools/software-tools/bbtools/bb-tools-user-guide/bbduk-guide/) by setting a minimum read length of 35 bp and a minimum Phred-quality score of 25. After trimming quality control, high-quality reads were aligned to the human genome reference (GRCh38) with STAR 2.7.3a (Dobin et al, 2013), while FeatureCounts 1.6.3 package (Liao et al, 2014) was used to assign reads to genes. Next, lowly expressed genes across one or more experimental conditions were filtered to eliminate the "uninformative" genes using HTSFilter v1.30.1 (Rau et al, 2013). Filtered gene data were further processed with the EdgeR package v3.32.1 (Robinson et al, 2010) to normalize (Trimmed Mean of M-7 values, TMM, method) the raw counts and perform differential gene expression analysis. Multiple testing correction was performed with the FDR method (Benjamini and Hochberg, 1995) and the significance level was set at FDR <0.05. Gene Ontology (GO) term enrichment was analyzed by performing hypergeometric tests11 for each individual term, and FDR correction was applied (FDR <0.05). Expression of Interspersed Repeat elements was quantified using SQuIRE 0.9.9.92 (https://github.com/wyang17/SQuIRE). SQuIRE provides locus-specific expression quantification along with subfamily-level expression estimates counting unambiguously mapped reads, as well as ambiguously mapped reads using an expectation–maximization (EM) algorithm (Yang et al, 2019). Briefly, reference genome and Repeatmasker annotation were downloaded from UCSC and prepared for the analysis with squire Fetch and squire Clean, respectively. High-quality reads were mapped against the reference genome with STAR using squire Map, and expression was quantified using squire Call. Differential expression analysis was performed with squire Call.

## DIA-MS analysis using TimsTOF MS

Total protein extracts were prepared with RIPA buffer (50 mMTris-cl pH 8.0, 5 mM EDTA, 150 mM NaCl, 15 mM MgCl$_2$, 1% NP-40, 1 mM PMSF, and 1X Protease Inhibitor Cocktail (PIC)). Further sonication step was included: (30 S ON, 30 S OFF, 10 cycles with Bioruptor). Equal 50 μg protein extracts were concentrated to 30 μl volume, then diluted in 8 M urea in 0.1 M Tris-HCl, followed by protein digestion with trypsin, according to the FASP protocol (Wiśniewski et al, 2009). After overnight digestion, the peptides were eluted from the filters with 25 mM ammonium bicarbonate buffer. The eluted peptides were processed in the desalting step using Sep-Pag C18 Column (waters) based on the manufacturer's instructions. Approximately 200 ng of peptide mixture per sample was analyzed using a timsTOF Pro 2 QTOF mass spectrometer coupled with a nanoElute liquid chromatography

system (Bruker Daltonik GmbH, Germany). The sample was injected directly into an RP-C18 Aurora emitter column (75 μm i.d. × 250 mm, 1.6 μm, 120 Å pore size) (Ion Optics, Australia) using a one-column separation method. An 80-min gradient was established using mobile phase A (0.1% FA in H2O) and mobile phase B (0.1% FA in Acetonitrile): 2–25% B for 60 min, 25–37% for 10 min, ramping 37 to 95% in 5 min, and maintaining 95% B for 5 min. The column temperature was set at 50 °C and the flow rate at 250 nl/min. The sample eluting from the separation column was introduced into the mass spectrometer via a CaptiveSpray nano-electrospray ion source (Bruker Daltonik GmbH) with an electrospray voltage of 1.6 kV. The ion source temperature was set to 180 °C and a dry gas of 3 l/min. The samples were analyzed using diaPASEF scheme (Meier et al, 2020b) consisting of 24 cycles including a total of 48 mass width windows (13 Da (m/z) from m/z 400 to 1000 and TIMS scan range from 0.63 to 1.35 Vs cm$^{-2}$ (1/K0). The collisional energy increased linearly from 20.01 eV at 0.6 (1/K0) to 52.00 eV at 1.35 Vs cm$^{-2}$ (1/K0). The scan range for MS and MS/MS spectra was set to 100–1700 m/z. TIMS ramping time and accumulation time were set to 100 ms. The diaPASEF data were analyzed by directDIA approach using Spectronaut software (version 14) following manufacture instructions. Up or down-regulated proteins were determined using the DEP (differential enrichment analysis of proteomics data) R package. Significant results (adjusted $p$ value <0.05) were subjected to Gene Ontology enrichment analysis with clusterProfiler R package.

## Western blot

Total protein extracts were prepared by lysing cells in extraction buffer (HEPES KOH [pH 8.5], NaCl 400 mM, EDTA 0.1 mM, EGTA 0.1 mM, DTT 1 mM, 1× protease inhibitor, SDS 1%). Proteins were separated by electrophoresis on BOLT 4–12% bis-tris polyacrylamide precast gels in MES buffer (Life Technologies) and transferred to a 0.2 μm nitrocellulose membrane. Non-specific signals were blocked with 5% Milk-PBS-Tween0.5% and the membrane hybridized overnight at 4 °C with primary and secondary antibodies diluted in a blocking buffer. Horseradish peroxidase-conjugated secondary antibodies were revealed with the ECL chemiluminescence kit (Amersham), and signals were detected using ChemiDoc (Bio-Rad). Primary antibodies: PKR (rabbit, Abcam, ab32052), cGAS (rabbit, Abcam, ab224144), LINE-1 ORF1 (mouse, Merck Millipore, MABC1152), EIF2a (mouse, Abcam, ab5369), phospho-EIF2 (rabbit, Abcam, ab32157), Histone H3 (rabbit, Merk, 06-755).

## Exosome isolation

Exosomes were isolated using Total Exosome Isolation Reagent (from cell culture media) (Thermo), following the manufacturer's instructions. The exosome pellet was resuspended in 50 μl of RIPA buffer and kept on ice for 30′, mixing every 5′. Samples were then sonicated (30″ ON/30″ OFF, 10 cycles), centrifuged at 13,000 × g for 20′ at 4 °C and collected as supernatants.

## Statistical analysis

Statistic tests used for data analysis are indicated in the figure legends.

## Data availability

The RNAseq transcriptomics data on human bone biopsies have been deposited to BioProject with the dataset identifier PRJNA764663. The RNAseq transcriptomics data on transfected osteoblasts have been deposited to GEO with the dataset identifier GSE201774 (enter token: ififogaehzoxlgh). The mass spectrometry proteomics data have been deposited to the ProteomeXchange Consortium via the PRIDE partner repository with the dataset identifier PXD051195.

The source data of this paper are collected in the following database record: biostudies:S-SCDT-10_1038-S44318-024-00143-z.

## Peer review information

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

## Acknowledgements

We acknowledge kaust Smarth Health Initiative (KSHI); the South-Eastern Norway Regional Health Authority; the Norwegian Osteoporosis Society; Anders Jahre Foundation for the Promotion of Science; the Lovisenberg Diaconal Hospital research fund; and the Research Council of Norway. We are grateful for support from: KAUST BAS/1/1037-01-01, Hevolution Foundation, the European Union project OSTEOGENE (no. FP6-502491); ZonMw VIDI 016.136.367 grant, for funding the creation of the RNA-Seq dataset of hip primary bone and Oslo University Hospital, Ullevål, project #29750104.

## Author contributions

**Arianna Mangiavacchi**: Conceptualization; Data curation; Formal analysis; Validation; Investigation; Visualization; Methodology; Writing—original draft; Project administration; Writing—review and editing. **Gabriele Morelli**: Conceptualization; Data curation; Formal analysis; Investigation; Methodology; Writing—review and editing. **Sjur Reppe**: Conceptualization; Resources; Data curation; Formal analysis; Investigation; Writing—review and editing. **Alfonso Saera- Vila**: Conceptualization; Data curation; Formal analysis; Investigation. **Peng Liu**: Conceptualization; Investigation; Methodology. **Benjamin Eggerschwiler**: Conceptualization; Resources; Formal analysis; Methodology. **Huoming Zhang**: Conceptualization; Methodology. **Dalila Bensaddek**: Conceptualization; Data curation; Methodology. **Elisa A Casanova**: Conceptualization; Resources; Methodology. **Carolina Medina-Gomez**: Conceptualization; Resources. **Vid Prijatelj**: Conceptualization; Resources. **Francesco Della Valle**: Conceptualization; Investigation. **Nazerke Atinbayeva**: Conceptualization; Investigation. **Juan Carlos Izpisua Belmonte**: Conceptualization; Writing—review and editing. **Fernando Rivadeneira**: Conceptualization; Resources. **Paolo Cinelli**: Conceptualization; Resources; Methodology; Writing—review and editing. **Kaare Morten Gautvik**: Conceptualization; Resources; Supervision; Investigation; Methodology; Writing—review and editing. **Valerio Orlando**: Conceptualization; Resources; Supervision; Funding acquisition; Investigation; Project administration; Writing—review and editing.

Source data underlying figure panels in this paper may have individual authorship assigned. Where available, figure panel/source data authorship is listed in the following database record: biostudies:S-SCDT-10_1038-S44318-024-00143-z.

## Disclosure and competing interests statement

The authors declare no competing interests.

