## [Peer Review File · The EMBO Journal]

LINE-1 RNA triggers matrix formation in bone cells via a PKR-mediated inflammatory response

Arianna Mangiavacchi, Gabriele Morelli, Sjur Reppe, Alfonso Saera Vila, Peng Liu, Benjamin Eggerschwiler, Huoming Zhang, Dalila Bensaddek, Elisa Casanova, Carolina Medina-Gomez, Vid Prijatelj, Francesco Della Valle, Nazerke Atinbayeva, Juan Carlos Izpisua Belmonte, Fernando Rivadeneira, Paolo Cinelli, Kaare Gautvik, and Valerio Orlando

Corresponding authors: Valerio Orlando (valerio.orlando@kaust.edu.sa) , Arianna Mangiavacchi (arianna.mangiavacchi@kaust.edu.sa)

Review Timeline:

Submission Date:	8th Aug 23
Editorial Decision:	18th Sep 23
Revision Received:	25th Mar 24
Editorial Decision:	10th May 24
Revision Received:	16th May 24
Accepted:	23rd May 24

Editor: Ieva Gailite

Transaction Report:

Dear Valerio,

Thank you for submitting your manuscript for consideration by the EMBO Journal. We have now received comments from three reviewers, which are included below for your information.

As you will see from the reports, all reviewers find the study of interest, while also pointing out a number of important aspects that would need to be addressed in the final manuscript before they can recommend acceptance of the manuscript. Based on the interest expressed in the reports, I would like to invite you to address the issues raised by the referees in a revised manuscript. I think it would be useful to discuss the revision in more detail via email or phone/videoconferencing - please let me know which option you prefer.

We generally allow three months as standard revision time, which can be extended to six months in the case of major revisions. As a matter of policy, competing manuscripts published during this period will not negatively impact on our assessment of the conceptual advance presented by your study. However, please contact me as soon as possible upon publication of any related work to discuss the appropriate course of action. Should you foresee a problem in meeting this deadline, please let us know in advance to discuss an extension.

When preparing your letter of response to the referees' comments, please bear in mind that this will form part of the Review Process File and will therefore be available online to the community. For more details on our Transparent Editorial Process, please visit our website: <https://www.embopress.org/page/journal/14602075/authorguide#transparentprocess>. Please also see the attached instructions for further guidelines on preparation of the revised manuscript.

Please feel free to contact me if you have any further questions regarding the revision. Thank you for the opportunity to consider your work for publication. I look forward to discussing your revision.

With best regards,

Ieva

We realize that it is difficult to revise to a specific deadline. In the interest of protecting the conceptual advance provided by the work, we recommend a revision within 3 months (17th Dec 2023). Please discuss the revision progress ahead of this time with the editor if you require more time to complete the revisions.

Referee #1:

The influence of retrotransposons, and especially LINE1s, in regulatory processes outside of harmful pathologies has become increasingly evident in recent years. Many of these L1 mediated mechanisms rely on the inflammatory response initiated by the accumulation of L1 cDNAs/hybrids in the cytoplasm, activating the cGAS/STING pathway. Mangiavacchi et. al highlight a new regulatory process influenced by L1 activation in bone repair, and further demonstrate the relevance of L1 activity in a wide variety of cellular processes and systems. The authors reference L1 dsRNAs as driving inflammation via PKR recognition and propose that this mechanism is potentially driving bone repair post-fracture. Given the literature demonstrating cGAS/STING activity as a driver of harmful inflammation, the authors take steps to demonstrate that neither L1 cDNA nor cGAS is implicated in their mechanism. Overall, the results presented by Mangiavacchi et. al show an exciting new mechanism driven by L1 RNAs that further bolster the importance of studying transposons in various cellular mechanisms across multiple tissues and cell types. The paper will be of interest to broad readership. The inclusion of several control experiments would strengthen the conclusions of this manuscript.

1. Using siRNA to target cGAS mRNA (and confirming this knockdown is successful) and demonstrating L1 RNA is still able to induce mineralization would further bolster their findings.
2. The data rely heavily on RNAseq analysis. The story would be stronger if the authors demonstrated L1 increases during fracture repair via another method. RT-qPCR targeting L1 mRNA would be a good way to quantify the increase in L1 expression between unfractured and healthy bone.
3. Immunofluorescent staining for ORF1/2p and/or DNA/RNA hybrids (S9.6 antibody) would serve as a convincing negative control to demonstrate the absence of L1 cDNAs and the L1-RNP complex in this process.

Vera Gorbunova

Referee #2:

Retrotransposons have been known as a major source of dsRNAs that trigger sterile immune responses, but their roles in bone metabolism are largely unknown. In the manuscript, the authors demonstrated that LINE-1 retrotransposon RNA acts as a bone anabolic stimulator during bone fracture healing. They found a transient upregulation of retrotransposons at inflammatory stage of bone fracture healing process in mice while there is a strong correlation between retrotransposon expression and bone anabolic demand and mineral density in human. L1 RNA treatment or the secretome of cells transfected with L1 RNA increased mineralization of BMSCs. Mechanistically, cytoplasmic L1 RNA, but not control RNA, activates PKR-mediated inflammation and bone formation in vitro. Overall, novel roles of LINE-1 RNA in extracellular matrix mineralization in osteoblasts, the soundness of the experimental approach, and potential in bone fracture therapy are appealing for publication in the EMBO Journal. However, the experiments were limited to in vitro cell culture and there are lack of detail mechanistic studies supporting PKR-mediated mineral production by L1 treatment and the secretome of cells transfected with L1 RNA.

Main concerns:

1. Figures 3 and 5: In addition to mineralization, osteoblast differentiation in L1 or the secretome-treated BSMCs should be assessed, such as CFU, alkaline phosphatase activity, and expression of osteoblast marker genes.
2. It is not clear how L1 RNA treatment upregulates mineralization of BMSCs. Does upregulation of chemokines result in mineral production? Authors need to clarify their contributions to mineralization.

3. In addition to regulation of protein synthesis via phosphorylation of E1F2a, PKR has been reported to regulate the WNT/beta-catenin signaling in osteoblasts by controlling GSK3-beta. Authors need to examine whether L1 RNA treatment enhances the WNT/beta-catenin signaling in BMSCs. Does the transcriptome analysis show upregulation of WNT signaling components in L1-treated cells? (Figure 4)
4. Page 6: the evidence is not strong enough to support the claim, "As shown (Figure 4D), their expression is already strongly reduced at day 10 and almost completely silenced by day 14, suggesting a transient dynamics of the earliest inflammatory response directly triggered by L1 RNA." To prove this, authors need to examine whether loss of L1 RNA can suppress inflammatory responses during bone fracture healing.
5. Since the PKR inhibitor C16 has off-target effects, authors need to confirm Figure 5 results using genetic deletion or knockdown of PKR.
6. Figure 5G: What components in the secretome of cells transfected with L1 RNA increase mineralization in BSMCs?

Miner Concerns:

1. Figure 3D and G legends: what are "images (central panels)"
2. Are LINE-1 retrotransposon RNA sequences conserved in mouse and human?
3. Figure 5G: is mineralized nodule formation different from extracellular mineralization?
4. There are multiple typos and spacing errors in the text, in particular, Materials and Methods section.

Referee #3:

The authors show that forced expression of LINE-1 retrotransposon RNA in mesenchymal stem cells (MSCs) from healthy or osteoporotic patients promotes osteogenic differentiation. The results, though very limited, suggest that these responses are mediated by protein kinase R (PKR). Strengths of this study is the novelty of the research topic in the context of bone pathophysiology, the high quality of the results, and their clear presentation. However, there are several concerns that need to be addressed as outlined below.

Major concerns:

- The authors are advised to restrain from unnecessary over-interpretations and overstatements such as: 1) femur is more anabolically active than the ilium (Figs. 1 and 2); 2) "L1 RNA delivery induces an inflammatory response characteristic of bone repair process": this is a huge stretch because these are in vitro results, though they correlate to some extent with published in vivo data; 3) the notion that PKR mediates anabolic response in vitro is also a stretch (simply state that it mediates osteoblast differentiation); 4) "in human, we found a significant correlation between repeats expression, microfracture-induced anabolic demand, and bone mineral density". Evidence of microfracture is not provided in this study.
- The abstract is incomprehensive, and the full manuscript needs to be deeply edited.
- If LINES are upregulated in conditions of "active bone anabolic processes" as claimed, why is L1 RNA not up-regulated during osteogenic differentiation of MSCs? What is the effect L1 inactivation on osteogenesis?
- References 60 and 61 are cited to make the point that Lamivudine 3TC and G140 inhibit the ORF2-mediated reverse transcription of L1 RNA and cGAS activation, respectively. These data need to be provided in the context of experimental conditions used in this study. Although L1 RNA induces PKR phosphorylation, off-target effects of C16 cannot be ruled out. Knocking down PKR would strengthen the conclusion.

Rev1

1. Using siRNA to target cGAS mRNA (and confirming this knockdown is successful) and demonstrating L1 RNA is still able to induce mineralization would further bolster their findings. We agree this is an important control and successfully performed the loss of function experiment.

We used a pool of cGAS targeting siRNAs at different concentrations (5nM, 25nM and 50nM) to reduce the expression of cGAS protein, as shown by western blot (Figure S6A). siRNAs were transfected at day 3 and WB analysis was done after 4 days (the same day L1 RNA was transfected). Successful cGAS knockdown did not affect L1 induced mineralization (Figure S6C).

2. The data rely heavily on RNAseq analysis. The story would be stronger if the authors demonstrated L1 increases during fracture repair via another method. RT-qPCR targeting L1 mRNA would be a good way to quantify the increase in L1 expression between unfractured and healthy bone.

We agree with Rev1 that a PCR validation regarding L1, and other repeats that we find to be differentially expressed the RNA-Seq, would further corroborate our data. Unfortunately, we do not have this material (RNA from fractured and unfractured mice) as we analyzed the global TEs expression (Fig. 1A and B) from a dataset published by another group (Coates et al, 2019). Furthermore, a comprehensive study (Everaert et al., 2017) showed a general high concordance between RT-qPCR and RNA-Seq with R2 values at ~0.93 when testing fold change correlations

between two sets of RNA for >10 000 genes. Thus, it is very unlikely that RT-qPCR analysis results would be different from those obtained by RNA-Seq.

3. Immunofluorescent staining for ORF1/2p and/or DNA/RNA hybrids (S9.6 antibody) would serve as a convincing negative control to demonstrate the absence of L1 cDNAs and the L1-RNP complex in this process.

ORF1p was undetectable by Western blot in differentiating osteoblasts, and in L1-RNA transfected osteoblasts, indicating that the exogenous L1-RNA is not translated and does not form RNP complexes with ORF1p (Figure S5). By immunofluorescence (not shown in the manuscript) we detected a nuclear signal for ORF1p, not colocalized with L1 RNA, and estimated to be unspecific. We found a significant colocalization between L1 RNA and S9.6 antibody (Figure S4). We cannot exclude that the antibody recognizes the dsRNA present in L1 RNA (Figure 5B), as previously reported (Hartono et al., 2018; Smolka et al., 2021). However, it is unlikely that L1 cDNA contributes to the observed phenotype as we proved that neither Lamivudine treatment (Figure 5A), cGAS inhibition (Figure 5A) or cGAS knockdown (Figure S6 A and C) affected L1-induced mineralization.

Figure S5: ORF1p expression in L1 transfected cells

Western Blot of ORF1 in untransfected (NT), RFP- and L1-transfected differentiating osteoblasts (OB). Tubulin is used as endogenous calibrator. Human breast cancer cell line (MCF7) extract is used as positive control.

Immunofluorescence of ORF1p (Figure not shown in the manuscript)

IF of ORF1p (green) in osteoblasts 48h after the transfection of cy5-coniugated L1-RNA (red).

Figure S4: Colocalization between exogenous L1 RNA and DNA:RNA hybrid IF signal

Immunofluorescence of DNA:RNA hybrids (green) in osteoblasts 24h after the transfection of cy5-coniugated L1 RNA (red).

Rev2

Main

concerns:

1. Figures 3 and 5: In addition to mineralization, osteoblast differentiation in L1 or the

secretome-treated BSMCs should be assessed, such as CFU, alkaline phosphatase activity, and expression of osteoblast marker genes.

We thank the reviewer for this suggestion. Therefore, we analyzed the expression of early and late osteogenic genes (Figure 3H, Figure S1), as well as the activity of alkaline phosphatase (ALPL) (Figure S2A) in L1 transfected cells. Osteoporotic patient-derived cells transfected with L1 RNA showed a higher expression of early marker genes, with a peak 48h after L1 transfection (day7) (Figure 3H). Bone sialoprotein (IBSP), a late osteogenic gene, showed a similar profile being almost 6 times higher expressed in L1- compared to RFP-treated cells at day 7 (Figure S1A). The expression of Osteocalcin (OCN), another late differentiation marker, increased significantly but only at day 13 (Figure S1). Moreover, although ALPL expression slightly increased at later time points (Figure S1A) its activity was not enhanced in L1 treated cells (Figure S2A). Therefore, we hypothesized that a different enzyme involved in hydroxyapatite formation might be affected by L1 RNA transfection thus contributing to the enhanced mineralization observed in L1 RNA overexpressing osteoblasts. Interestingly, we found that ENPP1 activity, which is known to inhibit osteoblast mineralization, was significantly reduced upon L1 RNA transfection (Figure S2B). We conclude that L1 RNA induces a unique mineralizing phenotype distinct from the well-known canonical differentiation mechanisms.

Figure 3H: qRT-PCR of early osteogenic genes

qRT-PCR of early osteogenic genes in RFP and L1 transfected osteoblasts at different time points of osteogenic differentiation. Expression level is normalized on day 5 (not transfected osteoblasts).

Figure S1: Timeline expression of late differentiation markers in L1 treated osteoblasts
 qRT-PCR of osteogenic genes in RFP and L1 transfected osteoblasts at different time points of osteogenic differentiation. Expression level is normalized on day 5 (not transfected osteoblasts).

Figure S2: ALPL and ENPP1 activity in L1 treated osteoblasts

A) Colorimetric ALPL activity assay of osteoblasts 5 days post-L1 transfection. B) ENPP1 activity of osteoblasts 6h, 5 days (d10) and 10 days (d15) after RFP and L1 transfection.

2. It is not clear how L1 RNA treatment upregulates mineralization of BMSCs. Does upregulation of chemokines result in mineral production? Authors need to clarify their contributions to mineralization.

We agree this is an interesting aspect. However, given the multitude of molecules involved, and complexity of molecular interaction, this depth of investigation goes far beyond the scope of the present work.

Moreover, the observation that inflammatory molecules induce osteoblast mineralization *in vitro* and *in vivo* has been previously reported (Croes et al., 2015; Croes et al., 2016; Li et al., 2016; Croes et al., 2017, Ferreira et al., 2013. Park et al., 2020), although the mechanism behind has not been elucidated yet.

However, following reviewer's request, we sought a suitable strategy to examine the possible contribution of chemokines to L1 induced-mineralization. Thus, we synthesized and used the pan chemokine inhibitor NR58-3.14.3 and its inactive derivative NR58-3.14.4 (Reckless et al., 2001; Miklos et al., 2009; Fernando et al., 2016). We performed the following experiment: differentiating osteoblasts (day 4) were incubated with increasing concentrations of the inhibitor (0,1-100 μ M) and then transfected with L1 RNA after 24h (day 5). We did not observe any changes in L1-induced mineralization. The lack of a positive control in the experiment makes it unfortunately impossible to conclude that chemokines do not contribute to *in vitro* L1-induced mineralization.

3. In addition to regulation of protein synthesis via phosphorylation of E1F2a, PKR has been reported to regulate the WNT/beta-catenin signaling in osteoblasts by controlling GSK3-beta. Authors need to examine whether L1 RNA treatment enhances the WNT/beta-catenin signaling in BMSCs. Does the transcriptome analysis show upregulation of WNT signaling components in L1-treated cells? (Figure 4)

We thank the reviewer for this important suggestion. We verified how the expression of WNT signaling components and final transcriptional targets changed upon L1 RNA transfection through Gene Set Enrichment Analysis (GSEA).

We tested several gene sets linked to WNT pathway and we found 6 and 2 gene sets that were down- and up-regulated 24h post L1 RNA transfection. However, none of the results obtained significance at FDR<25%.

4. Page 6: the evidence is not strong enough to support the claim, "As shown (Figure 4D), their expression is already strongly reduced at day 10 and almost completely silenced by day 14, suggesting a transient dynamics of the earliest inflammatory response directly triggered by L1 RNA." To prove this, authors need to examine whether loss of L1 RNA can suppress inflammatory responses during bone fracture healing.

We re-formulated the sentence in order to make it more clear it refers to our *in vitro* observations.

5. Since the PKR inhibitor C16 has off-target effects, authors need to confirm Figure 5 results using genetic deletion or knockdown of PKR.

We agree this is an important control and the loss of function experiment has been successfully performed. We used a pool of PKR targeting siRNAs at different concentrations (5nM, 25nM and 50nM) to reduce the expression of PKR protein, as shown by western blot (Figure S6A). siRNAs were transfected at day 3 and WB analysis was done after 4 days (the same day L1 RNA

was transfected). Consistently, PKR knockdown significantly reduced L1 RNA-induced mineralization (Figure S6B).

6. Figure 5G (now Figure 6A): What components in the secretome of cells transfected with L1 RNA increase mineralization in BSMCs?

We fully agree with Rev2 that this would be a breakthrough finding, particularly for the translational implication it may have. We have filed a patent on this regard and we are already moving forward with the deep characterization of the secretome (proteins, RNAs, metabolites) isolated from "L1 RNA-primed osteoblasts". To fully dissect the components in the secretome of cells transfected with L1 RNA leading to increased mineralization activity in osteoblasts is a major undertaking and lies outside the purpose of the present work. In the revised manuscript we however characterized the bulk- and exosome-derived proteomes from conditioned media of untreated, RFP- and L1-treated differentiating osteoblasts by MS (Figure 6). As discussed, "Differential expression analysis of MS data revealed a unique secretome profile (for both bulk and vesicular proteomes) of L1-primed osteoblasts compared to RFP-primed and untreated osteoblasts (Figure 6B). Gene Ontology (GO) analysis of differentially expressed proteins showed an enrichment of pro-inflammatory factors (i.e. interleukins and chemokines) involved in immune response and chemotactic migration of immune cells (Figure 6C and D), crucial processes for tissue repair mechanisms in vivo. Pro-inflammatory molecules are also a major constituent of the senescence-associated secretory phenotype (SASP), whose transient delivery supports cellular plasticity and tissue regeneration. Notably, the most enriched protein in L1-specific bulk secretome is interleukin 8 (IL-8), an inflammatory chemokine involved in several regenerative processes, such as skin wound healing, therapeutic angiogenesis after stroke and ischemia, and osteochondral bone repair. Tumor necrosis factor-inducible gene 6 (TSG6) is among the top 10 proteins enriched in the bulk secretome of L1-primed osteoblasts compared to RFP. TSG6 is an inflammatory factor with suggestive therapeutic effects in corneal wounds,

myocardial infarction, injured central nervous system, chronic liver damage, and intervertebral disc degeneration. Moreover, TSG6 induces autophagy influx both in vivo and in vitro. Another protein positively involved in autophagy and found specifically in the secretome of L1 primed osteoblasts is ATG7 (autophagy-related 7). Osteoblast-specific ATG7 conditional knockout in mice reduces bone mass at both developmental and adult age. Type 1 lysophosphatidic acid receptor (LPAR1) is also specifically secreted by L1-primed osteoblasts and positively involved in bone mineralization in vitro and in vivo.”

Future experiments will be necessary to study the detailed effects of each component in the context of bone mineralization/repair.

Figure 6

Fig. 6. L1-RNA-induced changes in osteoblast secretome

A) Left: Alizarin red staining on recipient OB 9 days after the delivery of conditioned media. Right: Microscope images of recipient OB 24h from the delivery of conditioned media. Mineralized nodules in OB receiving conditioned media from L1 treated OB are shown (orange arrows). B) Heat map of differentially expressed proteins in bulk secretome and exosomes derived from untransfected (NT), RFP- and L1-transfected osteoblasts. C) GO enrichment analysis of differentially expressed protein in the bulk secretome of L1 compared to RFP transfected osteoblasts. D) GO enrichment analysis of differentially expressed protein in the exosomes of L1 compared to RFP transfected osteoblasts.

Rev3

Major concerns:

- The authors are advised to restrain from unnecessary over-interpretations and overstatements such as:

1) femur is more anabolically active than the ilium (Figs. 1 and 2);

We appreciate the remark and have changed the text accordingly. Our arguments in relation to femur and ileum are based on well recognized studies indicating that mechanically loaded bone, like the femur, has a higher bone turnover rate (higher mineral, osteocalcin and IGF-I content, for example compared to other skeletal sites (ilium included) which experience less mechanical loading (e.g. Aerssens et al., 1997). Moreover, we previously reported that the ilium, subjected to low mechanical strain, shows a reduced number of bone related transcripts and their expression levels are markedly lower than in loaded skeletal sites, eg. spinal vertebra, certainly indicating a lower metabolic activity (Varanasi et al., 2010). Therefore, we modified the statement accordingly:

“Mechanically loaded bones, like the femur, which are subject to recurring microfractures that need to be repaired, are indicated to be more metabolically active than less loaded bones, like the iliac crest, as pointed out by previous transcriptome studies (Aerssens et al., 1997, Varanasi et al., 2010)”

2) "L1 RNA delivery induces an inflammatory response characteristic of bone repair process": this is a huge stretch because these are *in vitro* results, though they correlate to some extent with published *in vivo* data;

We re-formulated the statement: Indeed, our data show a significant (>50%) overlap between pathways induced during the early stage of *in vivo* fracture healing compared to our *in vitro* L1 delivery (Figure 4C). Therefore, more precisely, we wrote: *“L1 RNA delivery induces inflammatory pathways significantly overlapping those involved in bone fracture repair”*.

3) the notion that PKR mediates anabolic response *in vitro* is also a stretch (simply state that it mediates osteoblast differentiation);

As we found that L1 RNA induces a unique mineralizing phenotype not related to canonical differentiation mechanisms (Figure 3, Figure S1, Figure S2), we have throughout the text avoided the term “differentiation” and replaced “anabolic response” with “mineralization activity.”

4) "in human, we found a significant correlation between repeats expression, microfracture-induced anabolic demand, and bone mineral density". Evidence of microfracture is not provided in this study.

It is true that we do not provide data of microfracture in femur biopsies. However, it is well established medical fact that, in osteoporotic patients, low energy clinical fractures are always preceded by microfractures. In osteoporosis, fractures occur first in loaded bones, more frequently in the vertebra. Thus, clinical experience recognizes that the fracture risk is highest in mechanically loaded bones. Microfractures need to be repaired requiring higher bone remodeling and mineralization capacity. We have replaced "microfracture-induced" with "mechanical stress-induced" throughout the text.

5) The abstract is incomprehensive, and the full manuscript needs to be deeply edited.

The abstract has been rewritten and the manuscript thoroughly edited as suggested.

6) If LINES are upregulated in conditions of "active bone anabolic processes" as claimed, why is L1 RNA not up-regulated during osteogenic differentiation of MSCs? What is the effect L1 inactivation on osteogenesis?

To study the expression of L1 during all bone cell differentiation stages and the effect of L1 inactivation on osteogenesis in culture and *in vivo* would be definitely interesting, but is deemed outside the scope of the present work.

Our data show that L1-induced mineralization stimulates pathways distinct from those used in canonical osteoblastogenesis (Figure 3, Figure S1, Figure S2). We also included translational studies on stressed, compromised human bone and fractured mice where inflammatory reactions are a part of normal repair processes (Figure 1 and 2). We provide no information regarding a possible contribution of L1 reactivation in osteoblast differentiation. However, L1 is indeed involved in developmental processes, including neuronal differentiation, as previously demonstrated (Coufal et al., 2009).

7) References 60 and 61 are cited to make the point that Lamivudine 3TC and G140 inhibit the ORF2-mediated reverse transcription of L1 RNA and cGAS activation, respectively. These data need to be provided in the context of experimental conditions used in this study. Although L1 RNA induces PKR phosphorylation, off-target effects of C16 cannot be ruled out. Knocking down PKR would strengthen the conclusion.

We appreciate the comment, also given by Rev1. Indeed, this is an important control to further demonstrate that cGAS does not contribute to L1-induced mineralization. We performed the suggested experiment and included the results in the manuscript (Figure S6A and B). For details see also response to Rev 1 point 5.

References

Aerssens, J., Boonen, S., Joly, J. & Dequeker, J. Variations in trabecular bone composition with anatomical site and age: potential implications for bone quality assessment. *J. Endocrinol.* 155, 411–421 (1997).

Coates, B. A. et al. Transcriptional Profiling of Intramembranous and Endochondral Ossification after Fracture in Mice. *Bone* 127, 577 (2019).

Coleman M, Orvis A, Wu TY, Dacanay M, Merillat S, Ogle J, Baldessari A, Kretzer NM, Munson J, Boros-Rausch AJ, Shynlova O, Lye S, Rajagopal L, Adams Waldorf KM. A Broad Spectrum Chemokine Inhibitor Prevents Preterm Labor but Not Microbial Invasion of the Amniotic Cavity or Neonatal Morbidity in a Non-human Primate Model. *Front Immunol.* 2020 Apr 30;11:770. doi: 10.3389/fimmu.2020.00770. PMID: 32425945; PMCID: PMC7203489.

Coufal NG, Garcia-Perez JL, Peng GE, Yeo GW, Mu Y, Lovci MT, Morell M, O'Shea KS, Moran JV, Gage FH. L1 retrotransposition in human neural progenitor cells. *Nature.* 2009 Aug 27;460(7259):1127-31. doi: 10.1038/nature08248. Epub 2009 Aug 5. PMID: 19657334; PMCID: PMC2909034.

Croes, M. et al. Proinflammatory Mediators Enhance the Osteogenesis of Human Mesenchymal Stem Cells after Lineage Commitment. *PLoS One* 10, e0132781 (2015).

Croes, M. et al. Proinflammatory T cells and IL-17 stimulate osteoblast differentiation. *Bone* 84, 262–270 (2016).

Croes, M. et al. Inflammation-Induced Osteogenesis in a Rabbit Tibia Model. *Tissue Eng. Part C Methods* 23, 673–685 (2017).

Everaert C, Luybaert M, Maag JLV, Cheng QX, Dinger ME, Hellemans J, Mestdagh P. Benchmarking of RNA-sequencing analysis workflows using whole-transcriptome RT-qPCR expression data. *Sci Rep.* 2017 May 8;7(1):1559. doi: 10.1038/s41598-017-01617-3. PMID: 28484260; PMCID: PMC5431503.

Fernando, N., Natoli, R., Valter, K. *et al.* The broad-spectrum chemokine inhibitor NR58-3.14.3 modulates macrophage-mediated inflammation in the diseased retina. *J Neuroinflammation* 13, 47 (2016). <https://doi.org/10.1186/s12974-016-0514-x>

Ferreira E, Porter RM, Wehling N, O'Sullivan RP, Liu F, Boskey A, Estok DM, Harris MB, Vrahas MS, Evans CH, Wells JW. Inflammatory cytokines induce a unique mineralizing phenotype in mesenchymal stem cells derived from human bone marrow. *J Biol Chem.* 2013 Oct 11;288(41):29494-505. doi: 10.1074/jbc.M113.471268. Epub 2013 Aug 22. PMID: 23970554; PMCID: PMC3795248.

Hartono SR, Malapert A, Legros P, Bernard P, Chédin F, Vanoosthuyse V. The Affinity of the S9.6 Antibody for Double-Stranded RNAs Impacts the Accurate Mapping of R-Loops in Fission Yeast. *J Mol Biol.* 2018 Feb 2;430(3):272-284. doi: 10.1016/j.jmb.2017.12.016. Epub 2017 Dec 28. PMID: 29289567; PMCID: PMC5987549.

Li, C., Li, G., Liu, M., Zhou, T. & Zhou, H. Paracrine effect of inflammatory cytokine-activated bone marrow mesenchymal stem cells and its role in osteoblast function. *J. Biosci. Bioeng.* 121, 213–219 (2016).

Miklos S, Mueller G, Chang Y, Bouazzaoui A, Spacenko E, Schubert TEO, Grainger DJ, Holler E, Andreesen R, Hildebrandt GC. Preventive usage of broad spectrum chemokine inhibitor NR58-3.14.3 reduces the severity of pulmonary and hepatic graft-versus-host disease. *Int J Hematol.* 2009 Apr;89(3):383-397. doi: 10.1007/s12185-009-0272-y. Epub 2009 Mar 14. PMID: 19288173.

Park, J.-H.; Kang, Y.-H.; Hwang, S.-C.; Oh, S.H.; Byun, J.-H. Parthenolide Has Negative Effects on In Vitro Enhanced Osteogenic Phenotypes by Inflammatory Cytokine TNF- α via Inhibiting JNK Signaling. *Int. J. Mol. Sci.* 2020, 21, 5433.

Reckless J, Tatalick LM, Grainger DJ.. *Immunology.* 2001 Jun;103(2):244-54. doi: 10.1046/j.1365-2567.2001.01228.x. PMID: 11412312; PMCID: PMC1783230.

Smolka JA, Sanz LA, Hartono SR, Chédin F. Recognition of RNA by the S9.6 antibody creates pervasive artifacts when imaging RNA:DNA hybrids. *J Cell Biol.* 2021 Jun 7;220(6):e202004079. doi: 10.1083/jcb.202004079. PMID: 33830170; PMCID: PMC8040515.

Varanasi SS, Olstad OK, Swan DC, Sanderson P, Gautvik VT, Reppe S, Francis RM, Gautvik KM, Datta HK. Skeletal site-related variation in human trabecular bone transcriptome and signaling. *PLoS One.* 2010 May 18;5(5):e10692. doi: 10.1371/journal.pone.0010692. PMID: 20502692; PMCID: PMC2872667.

Dear Valerio,

Thank you for submitting a revised version of your manuscript. I sincerely apologise for the protracted assessment process due to delays in referee comment submission and the high number of submissions we receive at the moment.

Your study has now been seen by all original referees, who now find that their previous concerns have been addressed and recommend acceptance of the manuscript.

There now remain a few editorial points that need addressing before I can extend acceptance of the manuscript:

1. Emails sent to several of the authors bounced; please check the correctness of the email addresses for the authors Benjamin Eggerschwiler (benjamin.eggenschwiler@usz.ch), Nazerke Atinbayeva (atimbayeva@ie-freiburg.mpg.de), Elisa Casanova (Elisa.CasanovaZimmermann@usz.ch).
2. There is a mismatch between author name as indicated in the manuscript (Alfonso Saera) and in our online system (Alfonso Saera Vila), please check and correct.
3. Please define corresponding authors on the manuscript title page.
4. We require institutional email addresses for co-corresponding authors; it is currently missing for Kaare Morten Gautvik.
5. We are missing the ORCID iD for the co-corresponding author Kaare Morten Gautvik. In order to link the ORCID iD to the account in our manuscript tracking system, the author in question has to do the following:
 - Click the 'Modify Profile' link at the bottom of your homepage in our system.
 - On the next page you will see a box halfway down the page titled ORCID*. Below this box is red text reading 'To Register/Link to ORCID, click here'. Please follow that link: you will be taken to ORCID where you can log in to your account (or create an account if you don't have one)
 - You will then be asked to authorise Wiley to access your ORCID information. Once you have approved the linking, you will be brought back to our manuscript system.Unfortunately, we cannot do this linking on the author's behalf for security reasons.
6. Please upload the main and EV figures as individual production quality figure files in the .eps, .tif, or .jpg format (one file per figure).
7. Please make sure that the funding information is correct and identical both in the manuscript and our online system.
8. Please rename "Conflict of interest" section into "Disclosure and competing interests statement" (further info: <https://www.embopress.org/page/journal/14602075/authorguide#conflictsofinterest>).
9. Please move References before the Figure legend section.
10. Please update references according to The EMBO Journal style - where there are more than 10 authors on a paper, the first 10 should be listed, followed by 'et al.' Please see further information here: <https://www.embopress.org/page/journal/14602075/authorguide#referencesformat>
11. Please upload the Appendix file in PDF format and correct the nomenclature to "Appendix Figure S1" etc and "Appendix Table S1" etc. Please add a table of contents that includes page numbers.
12. In our standard source data check, we have noted unexplained duplicate values in source data for figure 4D. I have attached the corresponding files with the detected duplications labelled in orange. Please take a look and correct as needed. A brief explanation would be very helpful.
13. Our data editors have flagged the following issues in figure legends that need correcting:
 - Please provide the specific URLs for GSE201774 and PRJNA764663 datasets in the data availability statement.
 - Please define the annotated p values ****/****/**/* in the legend of figure 3d, g-h; as appropriate.
 - Please indicate the statistical test used for data analysis in the legends of figures 2d; 3d, g-h; 4d; 6c-d.
 - Please define the box plot in terms of minima, maxima, centre, bounds of box and whiskers, and percentile in the legend of figure 3f.
 - Please note define the error bars in the legends of figures 3b, d, g-h.
 - Please note that the scale bar is missing for figures 3d; 5b.
 - Please note that scale bar and its definition are missing for figures 3a, c; 6a.
 - Please define the white arrowheads in the legend of figure 3c.
14. Please submit a short 'Synopsis' to enhance discoverability of the manuscript. It consists of A) a short (1-2 sentences) summary of the findings and their significance, B) 3-4 bullet points highlighting key results.

Thank you again for giving us the chance to consider your manuscript for The EMBO Journal. I look forward to receiving the final version and your input on the source data issues.

With best wishes,

leva

We realize that it is difficult to revise to a specific deadline. In the interest of protecting the conceptual advance provided by the work, we recommend a revision within 3 months (8th Aug 2024). Please discuss the revision progress ahead of this time with the editor if you require more time to complete the revisions.

Referee #1:

The authors addressed all my concerns. I recommend publishing this highly significant study.

Referee #2:

All of the reviewers comments were well addressed. No more comments.

Referee #3:

The authors have satisfactorily addressed my concerns.

The authors addressed the remaining editorial issues.

Dear Valerio,

Thank you for providing the source data file and for your input on the textual edits. I will replace the blurb with your proposed version. I am now pleased to inform you that your manuscript has been accepted for publication in the EMBO Journal. Congratulations on a great study!

The typesetting process usually takes around two weeks and will also depend on your input on the proofs. Therefore, I think that approximately 3 weeks until online publication is a reasonable estimate.

If you have any questions, please do not hesitate to contact the Editorial Office. Thank you for your contribution to The EMBO Journal, and congratulations on a successful publication!

With best wishes,

Ieva

>>> Please note that it is The EMBO Journal policy for the transcript of the editorial process (containing referee reports and your response letter) to be published as an online supplement to each paper. If you do NOT want this, you will need to inform the Editorial Office via email immediately. More information is available here: https://www.embopress.org/transparent-process#Review_Process ogin page: <https://emboj.msubmit.net>